# Learning with Instance-Dependent Label Noise: A Sample Sieve Approach

**Hao Cheng**[*§]**, Zhaowei Zhu**[*†]**, Xingyu Li**[†]**, Yifei Gong**[§]**, Xing Sun**[§]**, and Yang Liu**[†]
[†]University of California, Santa Cruz, [§]Tencent YouTu Lab
`{zwzhu,xli279,yangliu}@ucsc.edu,`
`{louischeng,yifeigong,winfredsun}@tencent.com`

## Abstract

Human-annotated labels are often prone to noise, and the presence of such noise will degrade the performance of the resulting deep neural network (DNN) models. Much of the literature (with several recent exceptions) of learning with noisy labels focuses on the case when the label noise is independent of features. Practically, annotations errors tend to be instance-dependent and often depend on the difficulty levels of recognizing a certain task. Applying existing results from instance-independent settings would require a significant amount of estimation of noise rates. Therefore, providing theoretically rigorous solutions for learning with instance-dependent label noise remains a challenge. In this paper, we propose *CORES² (COnfidence REgularized Sample Sieve)*, which progressively sieves out corrupted examples. The implementation of CORES² does not require specifying noise rates and yet we are able to provide theoretical guarantees of CORES² in filtering out the corrupted examples. This high-quality sample sieve allows us to treat clean examples and the corrupted ones separately in training a DNN solution, and such a separation is shown to be advantageous in the instance-dependent noise setting. We demonstrate the performance of CORES² on CIFAR10 and CIFAR100 datasets with synthetic instance-dependent label noise and Clothing1M with real-world human noise. As of independent interests, our sample sieve provides a generic machinery for anatomizing noisy datasets and provides a flexible interface for various robust training techniques to further improve the performance. Code is available at `https://github.com/UCSC-REAL/cores`.

## 1 Introduction

Deep neural networks (DNNs) have gained popularity in a wide range of applications. The remarkable success of DNNs often relies on the availability of large-scale datasets. However, data annotation inevitably introduces label noise, and it is extremely expensive and time-consuming to clean up the corrupted labels. The existence of label noise can weaken the true correlation between features and labels as well as introducing artificial correlation patterns. Thus, mitigating the effects of noisy labels becomes a critical issue that needs careful treatment.

It is challenging to avoid overfitting to noisy labels, especially when the noise depends on both true labels $Y$ and features $X$. Unfortunately, this often tends to be the case where human annotations are prone to different levels of errors for tasks with varying difficulty levels. Recent work has also shown that the presence of instance-dependent noisy labels imposes additional challenges and cautions to training in this scenario (Liu, 2021). For such instance-dependent (or feature-dependent, instance-based) label noise settings, theory-supported works usually focus on loss-correction which requires estimating noise rates (Xia et al., 2020; Berthon et al., 2020). Recent work by Cheng et al. (2020) addresses the bounded instance-based noise by first learning the noisy distribution and then distilling examples according to some thresholds.[1] However, with a limited size of datasets, learning an accurate noisy distribution for each example is a non-trivial task. Additionally, the size and the quality of distilled examples are sensitive to the thresholds for distillation.

---

[*]Equal contributions in alphabetical ordering. Hao leads experiments and Zhaowei leads theories.
[†]Corresponding authors: Y. Liu and Z. Zhu `{yangliu,zwzhu}@ucsc.edu.`
[1]The proposed solution is primarily studied for the binary case in Cheng et al. (2020).

Departing from the above line of works, we design a sample sieve with theoretical guarantees to provide a high-quality splitting of clean and corrupted examples without the need to estimate noise rates. Instead of learning the noisy distributions or noise rates, we focus on learning the underlying clean distribution and design a regularization term to help improve the confidence of the learned classifier, which is proven to help safely sieve out corrupted examples. With the division between "clean" and "corrupted" examples, our training enjoys performance improvements by treating the clean examples (using standard loss) and the corrupted ones (using an unsupervised consistency loss) separately.

We summarize our main contributions: 1) We propose to train a classifier using a novel confidence regularization (CR) term and theoretically guarantee that, under mild assumptions, minimizing the confidence regularized cross-entropy (CE) loss on the instance-based noisy distribution is equivalent to minimizing the pure CE loss on the corresponding "unobservable" clean distribution. This classifier is also shown to be helpful for evaluating each example to build our sample sieve.2) We provide a theoretically sound sample sieve that simply compares the example's regularized loss with a closed-form threshold explicitly determined by predictions from the above trained model using our confidence regularized loss, without any extra estimates. 3) To the best of our knowledge, the proposed *CORES² (COnfidence REgularized Sample Sieve)* is the first method that is thoroughly studied for a multi-class classification problem, has theoretical guarantees to avoid overfitting to instance-dependent label noise, and provides high-quality division without knowing or estimating noise rates. 4) By *decoupling* the regularized loss into separate additive terms, we also provide a novel and promising mechanism for understanding and controlling the effects of general instance-dependent label noise. 5) CORES² achieves competitive performance on multiple datasets, including CIFAR-10, CIFAR-100, and Clothing1M, under different label noise settings.

**Other related works** In addition to recent works by Xia et al. (2020), Berthon et al. (2020), and Cheng et al. (2020), we briefly overview other most relevant references. Detailed related work is left to Appendix A. Making the loss function robust to label noise is important for building a robust machine learning model (Zhang et al., 2016). One popular direction is to perform *loss correction*, which first estimates transition matrix (Patrini et al., 2017; Vahdat, 2017; Xiao et al., 2015; Zhu et al., 2021b; Yao et al., 2020b), and then performs correction/reweighting via forward or backward propagation, or further revises the estimated transition matrix with controllable variations (Xia et al., 2019). The other line of work focuses on designing specific losses without estimating transition matrices (Natarajan et al., 2013; Xu et al., 2019; Liu & Guo, 2020; Wei & Liu, 2021). However, these works assume the label noise is instance-independent which limits their extension. Another approach is *sample selection* (Jiang et al., 2017; Han et al., 2018; Yu et al., 2019; Northcutt et al., 2019; Yao et al., 2020a; Wei et al., 2020; Zhang et al., 2020a), which selects the "small loss" examples as clean ones. However, we find this approach only works well on the instance-independent label noise. Approaches such as label correction (Veit et al., 2017; Li et al., 2017; Han et al., 2019) or semi-supervised learning (Li et al., 2020; Nguyen et al., 2019) also lack guarantees for the instance-based label noise.

## 2 CORES²: COnfidence REgularized Sample Sieve

Consider a classification problem on a set of $N$ training examples denoted by $D := \{(x_n, y_n)\}_{n \in [N]}$, where $[N] := \{1, 2, \cdots, N\}$ is the set of example indices. Examples $(x_n, y_n)$ are drawn according to random variables $(X, Y) \in \mathcal{X} \times \mathcal{Y}$ from a joint distribution $\mathcal{D}$. Let $\mathcal{D}_X$ and $\mathcal{D}_Y$ be the marginal distributions of $X$ and $Y$. The classification task aims to identify a classifier $f : \mathcal{X} \to \mathcal{Y}$ that maps $X$ to $Y$ accurately. One common approach is minimizing the empirical risk using DNNs with respect to the cross-entropy loss defined as $\ell(f(x), y) = -\ln(f_x[y])$, $y \in [K]$, where $f_x[y]$ denotes the $y$-th component of $f(x)$ and $K$ is the number of classes. In real-world applications, such as human-annotated images (Krizhevsky et al., 2012; Zhang et al., 2017) and medical diagnosis (Agarwal et al., 2016), the learner can only observe a set of noisy labels. For instance, human annotators may wrongly label some images containing cats as ones that contain dogs accidentally or irresponsibly. The label noise of each instance is characterized by a noise transition matrix $T(X)$, where each element $T_{ij}(X) := \mathbb{P}(\widetilde{Y} = j | Y = i, X)$. The corresponding noisy dataset[2] and distribution are denoted by $\widetilde{D} := \{(x_n, \tilde{y}_n)\}_{n \in [N]}$ and $\widetilde{\mathcal{D}}$. Let $\mathbb{1}(\cdot)$ be the indicator function taking

---

[2]In this paper, the noisy dataset refers to a dataset with noisy examples. A noisy example is either a clean example (whose label is true) or a corrupted example (whose label is wrong).

value 1 when the specified condition is satisfied and 0 otherwise. Similar to the goals in surrogate loss (Natarajan et al., 2013), $L_{\text{DMI}}$ (Xu et al., 2019) and peer loss (Liu & Guo, 2020), we aim to learn a classifier $f$ from the noisy distribution $\widetilde{\mathcal{D}}$ which also minimizes $\mathbb{P}(f(X) \neq Y), (X, Y) \sim \mathcal{D}$. Beyond their results, we attempt to propose *a theoretically sound approach addressing a general instance-based noise regime without knowing or estimating noise rates.*

## 2.1 CONFIDENCE REGULARIZATION

In this section, we present a new confidence regularizer (CR). Our design of the CR is mainly motivated by a recently proposed robust loss function called peer loss (Liu & Guo, 2020). For each example $(x_n, \tilde{y}_n)$, peer loss has the following form:

$$\ell_{\text{PL}}(f(x_n), \tilde{y}_n) := \ell(f(x_n), \tilde{y}_n) - \ell(f(x_{n_1}), \tilde{y}_{n_2}),$$

where $(x_{n_1}, \tilde{y}_{n_1})$ and $(x_{n_2}, \tilde{y}_{n_2})$ are two randomly sampled and paired peer examples (with replacement) for $n$. Let $X_{n_1}$ and $\widetilde{Y}_{n_2}$ be the corresponding random variables. Note $X_{n_1}, \widetilde{Y}_{n_2}$ are two independent and uniform random variables being each $x_{n'}, n' \in [N]$ and $\tilde{y}_{n'}, n' \in [N]$ with probability $\frac{1}{N}$ respectively: $\mathbb{P}(X_{n_1} = x_{n'}|\widetilde{D}) = \mathbb{P}(\widetilde{Y}_{n_2} = y_{n'}|\widetilde{D}) = \frac{1}{N}, \forall n' \in [N]$. Let $\mathcal{D}_{\widetilde{Y}|\widetilde{D}}$ be the distribution of $\widetilde{Y}_{n_2}$ given dataset $\widetilde{D}$. Peer loss then has the following equivalent form in expectation:

$$\frac{1}{N} \sum_{n \in [N]} \mathbb{E}_{X_{n_1}, \widetilde{Y}_{n_2}|\widetilde{D}}[\ell(f(x_n), \tilde{y}_n) - \ell(f(X_{n_1}), \widetilde{Y}_{n_2})]$$

$$= \frac{1}{N} \sum_{n \in [N]} \left[ \ell(f(x_n), \tilde{y}_n) - \sum_{n' \in [N]} \mathbb{P}(X_{n_1} = x_{n'}|\widetilde{D}) \mathbb{E}_{\mathcal{D}_{\widetilde{Y}|\widetilde{D}}}[\ell(f(x_{n'}), \widetilde{Y})] \right]$$

$$= \frac{1}{N} \sum_{n \in [N]} \left[ \ell(f(x_n), \tilde{y}_n) - \mathbb{E}_{\mathcal{D}_{\widetilde{Y}|\widetilde{D}}}[\ell(f(x_n), \widetilde{Y})] \right].$$

This result characterizes a new loss denoted by $\ell_{\text{CA}}$:

$$\ell_{\text{CA}}(f(x_n), \tilde{y}_n) := \ell(f(x_n), \tilde{y}_n) - \mathbb{E}_{\mathcal{D}_{\widetilde{Y}|\widetilde{D}}}[\ell(f(x_n), \widetilde{Y})]. \tag{1}$$

Though not studied rigorously by Liu & Guo (2020), we show, under conditions[3], $\ell_{\text{CA}}$ defined in Eqn. (1) encourages confident predictions[4] from $f$ by analyzing the gradients:

**Theorem 1.** *For $\ell_{CA}(\cdot)$, solutions satisfying $f_{x_n}[i] > 0, \forall i \in [K]$ are not locally optimal at $(x_n, \tilde{y}_n)$.*

See Appendix B.2 for the proof. Particularly, in binary cases, we have constraint $f(x_n)[0] + f(x_n)[1] = 1$. Following Theorem 1, we know minimizing $\ell_{\text{CA}}(f(x_n), \tilde{y}_n)$ w.r.t $f$ under this constraint leads to either $f(x_n)[0] \to 1$ or $f(x_n)[1] \to 1$, indicating confident predictions. Therefore, the addition of term $-\mathbb{E}_{\mathcal{D}_{\widetilde{Y}|\widetilde{D}}}[\ell(f(x_n), \widetilde{Y})]$ helps improve the confidence of the learned classifier. Inspired by the above observation, we define the following confidence regularizer:

$$\textbf{Confidence Regularizer:} \quad \ell_{\text{CR}}(f(x_n)) := -\beta \cdot \mathbb{E}_{\mathcal{D}_{\widetilde{Y}|\widetilde{D}}}[\ell(f(x_n), \widetilde{Y})],$$

where $\beta$ is positive and $\ell(\cdot)$ refers to the CE loss. The prior probability $\mathbb{P}(\widetilde{Y}|\widetilde{D})$ is counted directly from the noisy dataset. In the remaining of this paper, $\ell(\cdot)$ indicates the CE loss by default.

**Why are confident predictions important?** Intuitively, when model fits to the label noise, its predictions often become less confident, since the noise usually corrupts the signal encoded in the clean data. From this perspective, encouraging confident predictions plays against fitting to label noise. Compared to instance-independent noise, the difficulties in estimating the instance-dependent noise rates largely prevent us from applying existing techniques. In addition, as shown by Manwani & Sastry (2013), the 0-1 loss function is more robust to instance-based noise but hard to optimize with. To a certain degree, pushing confident predictions results in a differentiable loss function that approximates the 0-1 loss, and therefore restores the robustness property. Besides, as observed by Chatterjee (2020) and Zielinski et al. (2020), gradients from similar examples would reinforce each other. When the overall label information is dominantly informative that $T_{ii}(X) > T_{ij}(X)$, DNNs

---

[3]Detailed conditions for Theorem 1 are specified at the end of our main contents.

[4]Our observation can also help partially explain the robustness property of peer loss (Liu & Guo, 2020).

will receive more correct information statistically. Encouraging confident predictions would discourage the memorization of the noisy examples (makes it hard for noisy labels to reduce the confidence of predictions), and therefore further facilitate DNNs to learn the (clean) dominant information.

$\ell_{CR}$ **is NOT the entropy regularization** Entropy regularization (ER) is a popular choice for improving confidence of the trained classifiers in the literature (Tanaka et al., 2018; Yi & Wu, 2019). Given a particular prediction probability $p$ for a class, the ER term is based on the function $-p \ln p$, while our $\ell_{CR}$ is built on $\ln p$. Later we show $\ell_{CR}$ offers us favorable theoretical guarantees for training with instance-dependent label noise, while ER does not. In Appendix C.1, we present both theoretical and experimental evidences that $\ell_{CR}$ serves as a better regularizer compared to ER.

## 2.2 CONFIDENCE REGULARIZED SAMPLE SIEVE

Intuitively, label noise misleads the training thus sieving corrupted examples out of datasets is beneficial. Furthermore, label noise introduces high variance during training even with the existence of $\ell_{CR}$ (discussed in Section 3.3). Therefore, rather than accomplishing training solely with $\ell_{CR}$, we will first leverage its regularization power to design an efficient sample sieve. Similar to a general sieving process in physical words that compares the size of particles with the aperture of a sieve, we evaluate the "size" (quality, or a regularized loss) of examples and compare them with some to-be-specified thresholds, therefore the name sample sieve. In our formulation, the regularized loss $\ell(f(x_n), \tilde{y}_n) + \ell_{CR}(f(x_n))$ is employed to evaluate examples and $\alpha_n$ is used to specify thresholds. Specifically, we aim to solve the sample sieve problem in (2).

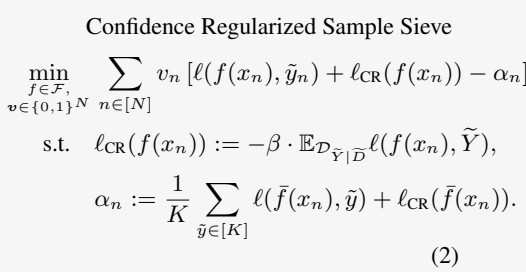

Confidence Regularized Sample Sieve

$$\min_{\substack{f \in \mathcal{F}, \\ \boldsymbol{v} \in \{0,1\}^N}} \sum_{n \in [N]} v_n \left[ \ell(f(x_n), \tilde{y}_n) + \ell_{CR}(f(x_n)) - \alpha_n \right]$$

s.t. $\ell_{CR}(f(x_n)) := -\beta \cdot \mathbb{E}_{\mathcal{D}_{\tilde{Y}|\tilde{D}}} \ell(f(x_n), \widetilde{Y}),$

$$\alpha_n := \frac{1}{K} \sum_{\tilde{y} \in [K]} \ell(\bar{f}(x_n), \tilde{y}) + \ell_{CR}(\bar{f}(x_n)).$$

(2)

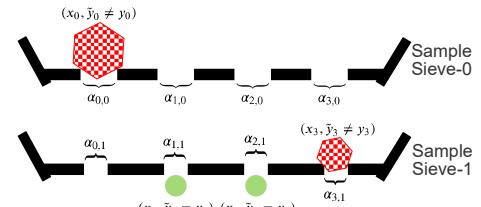

Figure 1: Dynamic sample sieves. Green circles are clean examples. Red hexagons are corrupted examples.

The crucial components in (2) are:

- $v_n \in \{0, 1\}$ indicates whether example $n$ is clean ($v_n = 1$) or not ($v_n = 0$);
- $\alpha_n$ (mimicking the aperture of a sieve) controls which example should be sieved out;
- $\bar{f}$ is a copy of $f$ and does not contribute to the back-propagation. $\mathcal{F}$ is the search space of $f$.

**Dynamic sample sieve** The problem in (2) is a combinatorial optimization which is hard to solve directly. A standard solution to (2) is to apply alternate search iteratively as follows:

- Starting at $t = 1, v_n^{(0)} = 1, \forall n \in [N]$.
- Confidence-regularized model update (at iteration-$t$):

$$f^{(t)} = \arg\min_{f \in \mathcal{F}} \sum_{n \in [N]} v_n^{(t-1)} \left[ \ell(f(x_n), \tilde{y}_n) + \ell_{CR}(f(x_n)) \right]; \quad (3)$$

- Sample sieve (at iteration-$t$):

$$v_n^{(t)} = \mathbb{1}(\ell(f^{(t)}(x_n), \tilde{y}_n) + \ell_{CR}(f^{(t)}(x_n)) < \alpha_{n,t}), \quad (4)$$

where $\alpha_{n,t} = \frac{1}{K} \sum_{\tilde{y} \in [K]} \ell(\bar{f}^{(t)}(x_n), \tilde{y}) + \ell_{CR}(\bar{f}^{(t)}(x_n))$, $f^{(t)}$ and $v^{(t)}$ refer to the specific classifier and weight at iteration-$t$. Note the values of $\ell_{CR}(\bar{f}^{(t)}(x_n))$ and $\ell_{CR}(f^{(t)}(x_n))$ are the same. We keep both terms to be consistent with the objective in Eq. (2). In DNNs, we usually update model $f$ with one or several epochs of data instead of completely solving (3).

Figure 1 illustrates the dynamic sample sieve, where the size of each example corresponds to the regularized loss and the aperture of a sieve is determined by $\alpha_{n,t}$. In each iteration-$t$, sample sieve-

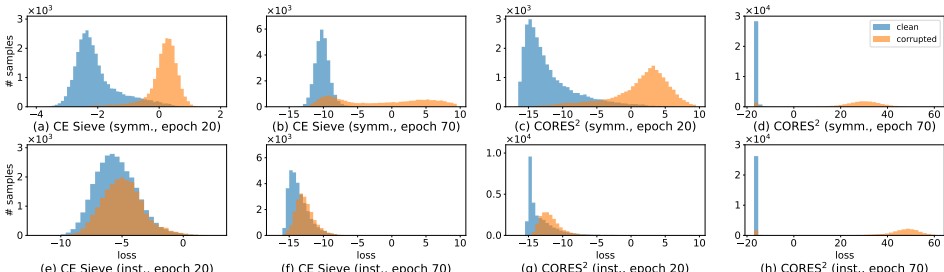

Figure 2: Loss distributions of training on CIFAR-10 with 40% symmetric noise (symm.) or 40% instance-based noise (inst.). The loss is given by $\ell(f^{(t)}(x_n), \tilde{y}_n) + \ell_{\text{CR}}(f^{(t)}(x_n)) - \alpha_{n,t}$ as (4). CE Sieve represents the dynamic sample sieve with standard cross-entropy loss (without CR).

$t$ "blocks" some corrupted examples by comparing a regularized example loss with a closed-form threshold $\alpha_{n,t}$, which can be immediately obtained given current model $\bar{f}^{(t)}$ and example $(x_n, \tilde{y}_n)$ (no extra estimation needed). In contrast, most sample selection works (Han et al., 2018; Yu et al., 2019; Wei et al., 2020) focus on controlling the number of the selected examples using an intuitive function where the overall noise rate may be required, or directly selecting examples by an empirically set threshold (Zhang & Sabuncu, 2018). Intuitively, the specially designed thresholds $\alpha_{n,t}$ for each example should be more accurate than a single threshold for the whole dataset. Besides, the goal of existing works is often to select clean examples while our sample sieve focuses on removing the corrupted ones. On a high level, we follow a different philosophy from these sample selection works. We coin our solution as COnfidence REgularized Sample Sieve (CORES²).

**More visualizations of the sample sieve** In addition to Figure 1, we visualize the superiority of our sample sieve with numerical results as Figure 2. The sieved dataset is in the form of two clusters of examples. Particularly, from Figure 2(b) and Figure 2(f), we observe that CE suffers from providing a good division of clean and corrupted examples due to overfitting in the final stage of training. On the other hand, with $\ell_{\text{CR}}$, there are two distinct clusters and can be separated by the threshold 0 as shown in Figure 2(d) and Figure 2(h). Comparing Figure 2(a)-2(d) with Figure 2(e)-2(h), we find the effect of instance-dependent noise on training is indeed different from the symmetric one, where the instance-dependent noise is more likely to cause overfitting.

## 3 THEORETICAL GUARANTEES OF CORES²

In this section, we theoretically show the advantages of CORES². The analyses focus on showing CORES² guarantees a quality division, i.e. $v_n = \mathbb{1}(y_n = \tilde{y}_n), \forall n$, with a properly set $\beta$. To show the effectiveness of this solution, we call a model prediction on $x_n$ is *better than random guess* if $f_{x_n}[y_n] > 1/K$, and call it *confident* if $f_{x_n}[y] \in \{0, 1\}, \forall y \in [K]$, where $y_n$ is the clean label and $y$ is an arbitrary label. The quality of sieving out corrupted examples is guaranteed in Theorem 2.

**Theorem 2.** *The sample sieve defined in (4) ensures that clean examples $(x_n, \tilde{y}_n = y_n)$ will not be identified as being corrupted if the model $f^{(t)}$'s prediction on $x_n$ is better than random guess.*

Theorem 2 informs us that our sample sieve can progressively and safely filter out corrupted examples, and therefore improves division quality, when the model prediction on each $x_n$ is better than random guess. The full proof is left to Appendix B.3. In the next section, we provide evidences that our trained model is guaranteed to achieve this requirement with sufficient examples.

### 3.1 DECOUPLING THE CONFIDENCE REGULARIZED LOSS

The discussion of performance guarantees of the sample sieve focuses on a general instance-based noise transition matrix $T(X)$, which can induce any specific noise regime such as symmetric noise and asymmetric noise (Kim et al., 2019; Li et al., 2020). Note the feature-independency was one critical assumption in state-of-the-art theoretically guaranteed noise-resistant literatures (Natarajan et al., 2013; Liu & Guo, 2020; Xu et al., 2019) while we *do not* require. Let $T_{ij} := \mathbb{E}_{\mathcal{D}|Y=i}[T_{ij}(X)], \forall i, j \in [K]$. Theorem 3 explicitly shows the contributions of clean examples, corrupted examples, and $\ell_{\text{CR}}$ during training. See Appendix B.1 for the proof.

**Theorem 3.** *(Main Theorem: Decoupling the Expected Regularized CE Loss) In expectation, the loss with $\ell_{CR}$ can be decoupled as three separate additive terms:*

$$\mathbb{E}_{\widetilde{\mathcal{D}}}\left[\ell(f(X),\widetilde{Y}) + \ell_{CR}(f(X))\right] = \overbrace{\underline{T} \cdot \mathbb{E}_{\mathcal{D}}[\ell(f(X),Y)]}^{\text{Term-1}} + \overbrace{\bar{\Delta} \cdot \mathbb{E}_{\mathcal{D}_\Delta}[\ell(f(X),Y)]}^{\text{Term-2}}$$
$$+ \underbrace{\sum_{j\in[K]}\sum_{i\in[K]}\mathbb{P}(Y=i)\mathbb{E}_{\mathcal{D}|Y=i}[(U_{ij}(X) - \beta\mathbb{P}(\widetilde{Y}=j))\ell(f(X),j)]}_{\text{Term-3}}, \tag{5}$$

*where* $\underline{T} := \min_{j\in[K]} T_{jj}$, $\bar{\Delta} := \sum_{j\in[K]}\Delta_j\mathbb{P}(Y=j)$, $\Delta_j := T_{jj} - \underline{T}$, $U_{ij}(X) = T_{ij}(X), \forall i \neq j$, $U_{jj}(X) = T_{jj}(X) - T_{jj}$, *and* $\mathbb{E}_{\mathcal{D}_\Delta}[\ell(f(X),Y)] := \mathbb{1}(\bar{\Delta} > 0)\sum_{j\in[K]}\frac{\Delta_j\mathbb{P}(Y=j)}{\bar{\Delta}}\mathbb{E}_{\mathcal{D}|Y=j}[\ell(f(X),j)]$.

Equation (5) provides a generic machinery for anatomizing noisy datasets, where we show the effects of instance-based label noise on the $\ell_{CR}$ regularized loss can be decoupled into three additive terms: **Term-1** reflects the expectation of CE on clean distribution $\mathcal{D}$, **Term-2** shifts the clean distribution by changing the prior probability of $Y$, and **Term-3** characterizes how the corrupted examples (represented by $U_{ij}(X)$) might mislead/mis-weight the loss, as well as the regularization ability of $\ell_{CR}$ (represented by $\beta\mathbb{P}(\widetilde{Y} = j)$). In addition to the design of sample sieve, this additive decoupling structure also provides a novel and promising perspective for understanding and controlling the effects of generic instance-dependent label noise.

## 3.2 GUARANTEES OF THE SAMPLE SIEVE

By decoupling the effects of instance-dependent noise into separate additive terms as shown in Theorem 3, we can further study under what conditions, minimizing the confidence regularized CE loss on the *(instance-dependent) noisy* distribution will be equivalent to minimizing the true loss incurred on the *clean* distribution, which is exactly encoded by Term-1. In other words, we would like to understand when Term-2 and Term-3 in (5) can be controlled not to disrupt the minimization of Term-1. Our next main result establishes this guarantee but will first need the following two assumptions.

**Assumption 1.** *($Y^* = Y$) Clean labels are Bayes optimal ($Y^* := \arg\max_{i\in[K]}\mathbb{P}(Y=i|X)$).*

**Assumption 2.** *(Informative datasets) The noise rate is bounded as $T_{ii}(X) - T_{ij}(X) > 0, \forall i \in [K], j \in [K], j \neq i, X \sim \mathcal{D}_X$.*

**Feasibility of assumptions:** 1) Note for many popular image datasets, e.g. CIFAR, the label of each feature is well-defined and the corresponding distribution is well-separated by human annotation. In this case, each feature $X$ only belongs to one particular class $Y$. Thus Assumption 1 is generally held in classification problems (Liu & Tao, 2015). Technically, this assumption could be relaxed. We use this assumption for clean presentations. 2) Assumption 2 shows the requirement of noise rates, i.e., for any feature $X$, a sufficient number of clean examples are necessary for dominant clean information. For example, we require $T_{ii}(X) - T_{ij}(X) > 0$ to ensure examples from class $i$ are informative (Liu & Chen, 2017).

Before formally presenting the noise-resistant property of training with $\ell_{CR}$, we discuss intuitions here. As discussed earlier in Section 2.1, our $\ell_{CR}$ regularizes the CE loss to generate/incentivize confident prediction, and thus is able to approximate the 0-1 loss to obtain its robustness property. More explicitly, from (5), $\ell_{CR}$ affects Term-3 with a scale parameter $\beta$. Recall that $U_{ij}(X) = T_{ij}(X), \forall i \neq j$, which is exactly the noise transition matrix. Although we have *no information* about this transition matrix, the confusion brought by $U_{ij}(X)$ can be canceled or reversed by a sufficiently large $\beta$ such that $U_{ij}(X) - \beta\mathbb{P}(\widetilde{Y} = j) \leq 0$. Intuitively, with an appropriate $\beta$, all the effects of $U_{ij}(X), i \neq j$ can be reversed, and we will get a negative loss punishing the classifier for predicting class-$j$ when the clean label is $i$. Formally, Theorem 4 shows the noise-resistant property of training with $\ell_{CR}$ and is proved in Appendix B.4.

**Theorem 4.** *(Robustness of the Confidence Regularized CE Loss) With Assumption 1 and 2, when*

$$\max_{i,j\in[K],X\sim\mathcal{D}_X}\frac{U_{ij}(X)}{\mathbb{P}(\widetilde{Y}=j)} \leq \beta \leq \min_{\mathbb{P}(\widetilde{Y}=i)>\mathbb{P}(\widetilde{Y}=j),X\sim\mathcal{D}_X}\frac{T_{ii}(X) - T_{ij}(X)}{\mathbb{P}(\widetilde{Y}=i) - \mathbb{P}(\widetilde{Y}=j)}, \tag{6}$$

*minimizing $\mathbb{E}_{\widetilde{\mathcal{D}}}[\ell(f(X),\widetilde{Y}) + \ell_{CR}(f(X))]$ is equivalent to minimizing $\mathbb{E}_{\mathcal{D}}[\ell(f(X),Y)]$.*

Theorem 4 shows a sufficient condition of $\beta$ for our confidence regularized CE loss to be robust to instance-dependent label noise. The bound on LHS ensures the confusion from label noise could be

canceled or reversed by the $\beta$ weighted confidence regularizer, and the RHS bound guarantees the model with the minimized regularized loss predicts the most frequent label in each feature w.p. 1.

Theorem 4 also provides guidelines for tuning $\beta$. Although we have no knowledge about $T_{ij}(X)$, we can roughly estimate the range of possible $\beta$. One possibly good setting of $\beta$ is linearly increasing with the number of classes, e.g. $\beta = 2$ for 10 classes and $\beta = 20$ for 100 classes.

With *infinite model capacity*, minimizing $\mathbb{E}_{\mathcal{D}}[\ell(f(X), Y)]$ returns the Bayes optimal classifier (since CE is a calibrated loss) which predicts on each $x_n$ better than random guess. Therefore, with a sufficient number of examples, minimizing $\mathbb{E}_{\widetilde{\mathcal{D}}}[\ell(f(X), \widetilde{Y}) + \ell_{CR}(f(X))]$ will also return a model that predicts better than random guess, then satisfying the condition required in Theorem 2 to guarantee the quality of sieved examples. Further, since the Bayes optimal classifier always predicts clean labels confidently when Assumption 1 holds, Theorem 4 also guarantees confident predictions. With such predictions, the sample sieve in (4) will achieve $100\%$ precision on both clean and corrupted examples. This guaranteed division is summarized in Corollary 1:

**Corollary 1.** *When conditions in Theorem 4 hold, with infinite model capacity and sufficiently many examples, CORES$^2$ achieves $v_n = \mathbb{1}(y_n = \tilde{y}_n), \forall n \in [N]$, i.e., all the sieved clean examples are effectively clean.*

### 3.3 TRAINING WITH SIEVED SAMPLES

We discuss the necessity of a dynamic sample sieve in this subsection. Despite the strong guarantee in expectation as shown Theorem 4, performing direct Empirical Risk Minimization (ERM) of the regularized loss is likely to return a sub-optimal solution. Although Theorem 4 guarantees the equivalence of minimizing two first-order statistics, their *second-order statistics* are also important for estimating the expectation when examples are finite. Intuitively, Term-1 $\underline{T} \cdot \mathbb{E}_{\mathcal{D}}[\ell(f(X), Y)]$ primarily helps distinguish a good classifier from a bad one on the clean distribution. The existence of the leading constant $\underline{T}$ reduces the power of the above discrimination, as effectively the gap between the expected losses become smaller as noise increases ($\underline{T}$ will decrease). Therefore we would require more examples to recognize the better model. Equivalently, the variance of the selection becomes larger. In Appendix C.2, we also offer an explanation from the variance's perspective. For some instances with extreme label noise, the $\beta$ satisfying Eqn. (6) in Theorem 4 may not exist. In such case, these instances cannot be properly used and other auxiliary techniques are necessary (e.g., sample pruning).

Sieving out the corrupted examples from the clean ones allows us a couple of better solutions. First, we can focus on performing ERM using these sieved clean examples only. We derive the risk bound for training with these clean examples in Appendix C.3. Secondly, leveraging the sample sieve to distinguish clean examples from corrupted ones provides a flexible interface for various robust training techniques such that the performance can be further improved. For example, semi-supervised learning techniques can be applied (see section 4 for more details).

## 4 EXPERIMENTS

Now we present experimental evidences of how CORES$^2$ works. [5]

**Datasets:** CORES$^2$ is evaluated on three benchmark datasets: CIFAR-10, CIFAR-100 (Krizhevsky et al., 2009) and Clothing1M (Xiao et al., 2015). Following the convention from Xu et al. (2019), we use ResNet34 for CIFAR-10 and CIFAR-100 and ResNet50 for Clothing1M.

**Noise type:** We experiment with three types of label noise: symmetric, asymmetric and instance-dependent label noise. Symmetric noise is generated by randomly flipping a true label to the other possible labels w.p. $\varepsilon$ (Kim et al., 2019), where $\varepsilon$ is called the noise rate. Asymmetric noise is generated by flipping the true label to the next class (*i.e.*, label $i \rightarrow i+1, \mod K$) w.p. $\varepsilon$. Instance-dependent label noise is a more challenging setting and we generate instance-dependent label noise following the method from Xia et al. (2020) (See Appendix D.3 for details). In expectation, the noise rate $\varepsilon$ for all noise regimes is the overall ratio of corrupted examples in the whole dataset.

**Consistency training after the sample sieve:** Let $\tau$ be the last iteration of CORES$^2$. Define $L(\tau) := \{n | n \in [N], v_n^{(\tau)} = 1\}$, $H(\tau) := \{n | n \in [N], v_n^{(\tau)} = 0\}$, $\widetilde{D}_{L(\tau)} := \{(x_n, \tilde{y}_n) : n \in$

---

[5]The logarithmic function in $\ell_{CR}$ is adapted to $\ln(f_x[y] + 10^{-8})$ for numerical stability.

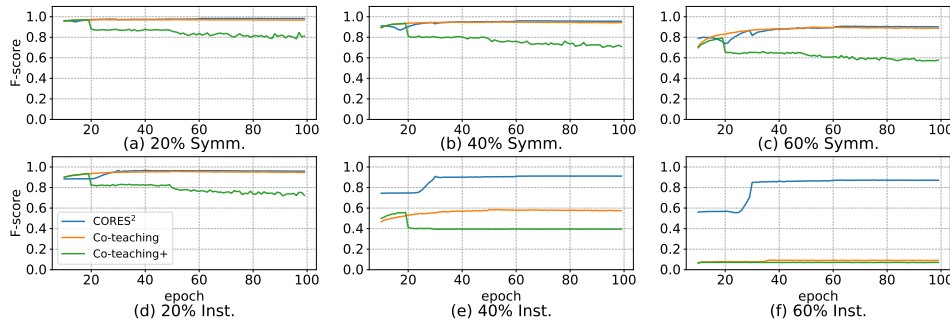

Figure 3: F-score comparisons on CIFAR10 under symmetric (Symm.) and instance-based (Inst.) label noise. $\text{F-score} := \frac{2 \cdot \text{Pre} \cdot \text{Re}}{\text{Pre} + \text{Re}}$, where $\text{Pre} := \frac{\sum_{n \in [N]} \mathbb{1}(v_n=1, y_n=\tilde{y}_n)}{\sum_{n \in [N]} \mathbb{1}(v_n=1)}$, and $\text{Re} := \frac{\sum_{n \in [N]} \mathbb{1}(v_n=1, y_n=\tilde{y}_n)}{\sum_{n \in [N]} \mathbb{1}(y_n=\tilde{y}_n)}$.

$L(\tau)\}$, $\widetilde{D}_{H(\tau)} := \{(x_n, \tilde{y}_n) : n \in H(\tau)\}$. Thus $\widetilde{D}_{L(\tau)}$ is sieved as clean examples and $\widetilde{D}_{H(\tau)}$ is filtered out as corrupted ones. Examples $(x_n, \tilde{y}_n) \in \widetilde{D}_{L(\tau)}$ lead the training direction using the CE loss as $\sum_{n \in L(\tau)} \ell(f(x_n), \tilde{y}_n)$. Noting the labels in $\widetilde{D}_{H(\tau)}$ are supposed to be corrupted and can distract the training, we simply drop them. On the other hand, feature information of these examples encodes useful information that we can further leverage to improve the generalization ability of models. There are different ways to use this unsupervised information, in this paper, we chose to minimize the KL-divergence between predictions on the original feature and the augmented feature to make predictions consistent. This is a common option as chosen by Li et al. (2019), Xie et al. (2019), and Zhang et al. (2020b). The consistency loss function in epoch-$t$ is $\sum_{n \in H(\tau)} \ell_{\text{KL}}(f(x_n), \bar{f}^{(t)}(x_{n,t}))$, where $\bar{f}^{(t)}$ is a copy of the DNN at the beginning of epoch-$t$ but without gradients. Summing the classification and consistency loss yields the total loss. See Appendix D.1 for an illustration.

**Other alternatives:** Checking the consistency of noisy predictions is only one possible way to leverage the additional information after sample sieves. Our basic idea of first sieving the dataset and then treating corrupted examples differently from clean ones admits other alternatives. There are many other possible designs after sample sieves, e.g., estimating transition matrix using sieved examples then applying loss-correction (Patrini et al., 2017; Vahdat, 2017; Xiao et al., 2015), making the consistency loss as another regularization term and retraining the model (Zhang et al., 2020b), correcting the sample selection bias in clean examples and retraining (Cheng et al., 2020; Fang et al., 2020), or relabeling those corrupted examples and retraining, etc. Additionally, clustering methods on the feature space (Han et al., 2019; Luo et al., 2020) or high-order information (Zhu et al., 2021a) can also be exploited along with the dynamic sample sieve. Besides, the current structure is ready to include other techniques such as mixup (Zhang et al., 2018).

**Quality of our sample sieve:** Figure 3 shows the F-scores of sieved clean examples with training epochs on the symmetric and the instance-based label noise. F-score quantifies the quality of the sample sieve by the harmonic mean of precision (ratio of actual cleans examples in sieved clean ones) and recall (ratio of sieved cleans examples in actual clean ones). We compare CORES$^2$ with Co-teaching and Co-teaching+. Note the F-scores of CORES$^2$ and Co-teaching are consistently high on the symmetric noise, while CORES$^2$ achieves higher performance on the challenging instance-based label noise, especially with the $60\%$ noise rate where the other two methods have low F-scores.

**Experiments on CIFAR-10, CIFAR-100 and Clothing1M:** In this section, we compare CORES$^2$ with several state-of-the-art methods on CIFAR-10 and CIFAR-100 under instance-based, symmetric and asymmetric label noise settings, which is shown on Table 1 and Table 2. CORES$^{2\star}$ denotes that we apply consistency training on the corrupted examples after the sample sieve. For a fair comparison, all the methods use ResNet-34 as the backbone. By comparing the performance of CE on the symmetric and the instance-based label noise, we note the instance-based label noise is a more challenging setting. Even though some methods (e.g., $L_{\text{DMI}}$) behaves well on symmetric and asymmetric label noise, they may reach low test accuracies on the instance-based label noise, especially when the noise rate is high or the dataset is more complex. However, CORES$^2$ consistently works well on the instance-based label noise and adding the consistency training gets better results. Table 3 verifies CORES$^2$ on Clothing1M, a dataset with real human label noise. Compared to the other

Table 1: Comparison of test accuracies on clean datasets under instance-based label noise.

| Method | Inst. CIFAR10 | | | Inst. CIFAR100 | | |
|---|---|---|---|---|---|---|
| | $\varepsilon = 0.2$ | $\varepsilon = 0.4$ | $\varepsilon = 0.6$ | $\varepsilon = 0.2$ | $\varepsilon = 0.4$ | $\varepsilon = 0.6$ |
| Cross Entropy | 87.16 | 75.16 | 44.64 | 58.72 | 41.14 | 25.29 |
| Forward $T$ (Patrini et al., 2017) | 88.08 | 82.67 | 41.57 | 58.95 | 41.68 | 22.83 |
| $L_{\text{DMI}}$ (Xu et al., 2019) | 88.80 | 82.70 | 70.54 | 58.66 | 41.77 | 28.00 |
| $L_q$ (Zhang & Sabuncu, 2018) | 86.45 | 69.02 | 32.94 | 58.18 | 40.32 | 23.13 |
| SCE (Wang et al., 2019) | 89.11 | 72.04 | 44.83 | 59.87 | 41.76 | 23.41 |
| Co-teaching (Han et al., 2018) | 88.66 | 69.50 | 34.61 | 43.03 | 23.13 | 7.07 |
| Co-teaching+ (Yu et al., 2019) | 89.04 | 69.15 | 33.33 | 41.84 | 24.40 | 8.74 |
| JoCoR (Wei et al., 2020) | 88.71 | 68.97 | 30.27 | 44.28 | 22.77 | 7.54 |
| Peer Loss (Liu & Guo, 2020) | 89.33 | 81.09 | 73.73 | 59.92 | 45.76 | 33.61 |
| CORES$^2$ | **89.50** | **82.84** | **79.66** | **61.25** | **47.81** | **37.85** |
| CORES$^{2\star}$ | **95.42** | **88.45** | **85.53** | **72.91** | **70.66** | **63.08** |

Table 2: Comparison of test accuracies on clean datasets under symmetric/asymmetric label noise.

| Method | Symm. CIFAR10 | | Asymm. CIFAR10 | | Symm. CIFAR100 | | Asymm. CIFAR100 | |
|---|---|---|---|---|---|---|---|---|
| | $\varepsilon = 0.4$ | $\varepsilon = 0.6$ | $\varepsilon = 0.2$ | $\varepsilon = 0.3$ | $\varepsilon = 0.4$ | $\varepsilon = 0.6$ | $\varepsilon = 0.2$ | $\varepsilon = 0.3$ |
| Cross Entropy | 81.88 | 74.14 | 88.59 | 86.14 | 48.20 | 37.41 | 59.20 | 51.40 |
| MAE (Ghosh et al., 2017) | 61.63 | 41.98 | 59.67 | 57.62 | 7.68 | 6.45 | 11.16 | 8.97 |
| Forward $T$ (Patrini et al., 2017) | 83.27 | 75.34 | 89.42 | 88.25 | 53.04 | 41.59 | 64.86 | 64.72 |
| $L_q$ (Zhang & Sabuncu, 2018) | 87.13 | 82.54 | 89.33 | 85.45 | 61.77 | 53.16 | 66.59 | 61.45 |
| $L_{\text{DMI}}$ (Xu et al., 2019) | 83.04 | 76.51 | 89.04 | 87.88 | 52.32 | 40.00 | 60.04 | 52.82 |
| NLNL (Kim et al., 2019) | 92.43 | 88.32 | 93.35 | 91.80 | 66.39 | 56.51 | 63.12 | 54.87 |
| SELF (Nguyen et al., 2019) | 91.13 | - | 93.75 | 92.42 | 66.71 | - | 70.53 | 65.09 |
| CORES$^{2\star}$ | **93.76** | **89.78** | **95.18** | **94.67** | **72.22** | **59.16** | **75.19** | **73.81** |

Table 3: The best epoch (clean) test accuracy for each method on Clothing1M.

| Method | CE (Baseline) | Forward $T$ (Patrini et al., 2017) | Co-teaching (Han et al., 2018) | JoCoR (Wei et al., 2020) | $L_{\text{DMI}}$ (Xu et al., 2019) | PTD-R-V (Xia et al., 2020) | CORES$^2$ (our) |
|---|---|---|---|---|---|---|---|
| Acc. | 68.94 | 70.83 | 69.21 | 70.30 | 72.46 | 71.67 | **73.24** |

approaches, CORES$^2$ also works fairly well on the Clothing1M dataset. See more experiments in Appendix D. We also provide source codes with detailed instructions in supplementary materials.

## 5 CONCLUSIONS

This paper introduces CORES$^2$, a sample sieve that is guaranteed to be robust to general instance-dependent label noise and sieve out corrupted examples, but without using explicit knowledge of the noise rates of labels. The analysis of CORES$^2$ assumed that the Bayes optimal labels are the same as clean labels. Future directions of this work include extensions to more general cases where the Bayes optimal labels may differ from clean labels. We are also interested in exploring different possible designs of robust training with sieved examples.

**Acknowledgement** This work is partially supported by the National Science Foundation (NSF) under grant IIS-2007951 and the Office of Naval Research under grant N00014-20-1-22.

## CONDITIONS REQUIRED FOR THEOREM 1

Theorem 1 holds based on the following three assumptions:

A1. The model capacity is infinite (i.e., it can realize arbitrary variation).

A2. The model is updated using the gradient descent algorithm (i.e. updates follow the direction of decreasing $\mathbb{E}_{\mathcal{D}}\left[\ell(f(X), Y)\right] - \mathbb{E}_{\mathcal{D}_Y}\left[\mathbb{E}_{\mathcal{D}_X}\left[\ell(f(X), Y)\right]\right]$).

A3. The derivative of network function $\frac{\partial f(x;w)}{\partial w_i}$ is smooth (i.e. the network function has no singular point), where $w_i$'s are model parameters.

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

APPENDIX

The appendices are organized as follows. Section A presents the full version of related works. Section B details the proofs for our theorems. Section C supplements other necessary evidences to justify CORES$^2$. Section D shows more experimental details and results.

## A    FULL VERSION OF RELATED WORKS

Learning with noisy labels has observed exponentially growing interests. Since the traditional cross-entropy (CE) loss has been proved to easily overfit noisy labels (Zhang et al., 2016), researchers try to design different loss functions to handle this problem. There were two main perspectives on *designing loss functions*. Considering the fact that outputs of logarithm functions in the CE loss grow explosively when the prediction $f(x)$ approaches zero, some researchers tried to design bounded loss functions (Amid et al., 2019; Wang et al., 2019; Gong et al., 2018; Ghosh et al., 2017). To avoid relying on fine-tuning of hyper-parameters in loss functions, a meta-learning method was proposed bt Shu et al. (2020) to combine the above four loss functions together. However, simply considering loss function values without discussing the noise type and the corresponding statistics could not be noise-tolerant as defined by Manwani & Sastry (2013). As a complementary, others started from noise types and tried to design noise-tolerant loss functions. Based on the assumption that label noise only depends on the true class (a.k.a. feature-independent or label-dependent), an unbiased loss function called surrogate loss (Natarajan et al., 2013), an information-based loss function called $L_{\text{DMI}}$ (Xu et al., 2019), and a new family of loss functions to punish agreements between classifiers and noisy datasets called peer loss (Liu & Guo, 2020) were proposed. They proved theoretically that training DNNs using their loss functions on feature-independent noisy datasets was equivalent to training CE on the corresponding unobservable clean datasets. However, surrogate loss focused on the binary classifications and required knowing noise rates. $L_{\text{DMI}}$ and peer loss does not require knowing noise rates while $L_{\text{DMI}}$ may not be easy for extension and multi-class classification of peer loss requires particular transition matrices.

The *correction* approach is also popular in handling label noise. Previous works (Patrini et al., 2017; Vahdat, 2017; Xiao et al., 2015) assumed the feature-independent noise transition matrix was given or could be estimated and attempted to use it to correct loss functions. For example, Patrini et al. (2017) first estimated the noise transition matrix and then relied on it to correct forward or backward propagation during training. However, without a set of clean examples, the noise transition matrix could be hard to estimate correctly. Instead of correcting loss functions, some methods directly corrected labels (Veit et al., 2017; Li et al., 2017; Han et al., 2019), whereas it might introduce extra noise and damage useful information. Recent works (Xia et al., 2020; Berthon et al., 2020) extended loss-correction from the limited feature-independent label noise to part-dependent or a more general instance-dependent noise regime while they relied heavily on the noise rate estimation.

*Sample selection* (Jiang et al., 2017; Han et al., 2018; Yu et al., 2019; Yao et al., 2020a; Wei et al., 2020) mainly focused on exploiting the memorization of DNNs and treating the "small loss" examples as clean ones, while they only focused on feature-independent label noise. Cheng et al. (2020) tried to distill some examples relying on the predictions using the surrogate loss function (Natarajan et al., 2013). Note estimating noise rates are necessary for both applying surrogate loss and determining the threshold for distillation. The sample selection methods could be implemented with some semi-supervised learning techniques to improve the performance, where the corrupted examples were treated as unlabeled data (Li et al., 2020; Nguyen et al., 2019). However, the training mechanisms of these methods were still based on the CE loss, which could not be guaranteed to avoid overfitting to label noise.

## B    PROOF FOR THEOREMS

In this section, we firstly present the proof for Theorem 3 (our main theorem) in Section B.1, which provides a generic machinery for anatomizing noisy datasets. Then we will respectively prove Theorem 1 in Section B.2, Theorem 2 in Section B.3, and Theorem 4 in Section B.4 according to the order they appear.

## B.1 PROOF FOR THEOREM 3

**Theorem 3.** *(Main Theorem: Decoupling the Expected Regularized CE Loss) In expectation, the loss with $\ell_{CR}$ can be decoupled as three separate additive terms:*

$$
\mathbb{E}_{\widetilde{\mathcal{D}}}\left[\ell(f(X),\widetilde{Y}) + \ell_{CR}(f(X))\right] = \overbrace{\underline{T}\cdot\mathbb{E}_{\mathcal{D}}[\ell(f(X),Y)]}^{\text{Term-1}} + \overbrace{\bar{\Delta}\cdot\mathbb{E}_{\mathcal{D}_\Delta}[\ell(f(X),Y)]}^{\text{Term-2}}
$$
$$
+ \underbrace{\sum_{j\in[K]}\sum_{i\in[K]}\mathbb{P}(Y=i)\mathbb{E}_{\mathcal{D}|Y=i}[(U_{ij}(X) - \beta\mathbb{P}(\widetilde{Y}=j))\ell(f(X),j)]}_{\text{Term-3}}, \tag{7}
$$

*where $\underline{T} := \min_{j\in[K]} T_{jj}$, $\bar{\Delta} := \sum_{j\in[K]}\Delta_j\mathbb{P}(Y=j)$, $\Delta_j := T_{jj} - \underline{T}$, $U_{ij}(X) = T_{ij}(X), \forall i \neq j, U_{jj}(X) = T_{jj}(X) - T_{jj}$, and $\mathbb{E}_{\mathcal{D}_\Delta}[\ell(f(X),Y)] := \mathbb{1}(\bar{\Delta} > 0)\sum_{j\in[K]}\frac{\Delta_j\mathbb{P}(Y=j)}{\bar{\Delta}}\mathbb{E}_{\mathcal{D}|Y=j}[\ell(f(X),j)]$.*

*Proof.* The expected form of traditional CE loss on noisy distribution $\widetilde{\mathcal{D}}$ can be written as

$$
\mathbb{E}_{\widetilde{\mathcal{D}}}[\ell(f(X),\widetilde{Y})]
$$
$$
= \sum_{j\in[K]}\sum_{i\in[K]}\mathbb{P}(Y=i)\mathbb{E}_{\mathcal{D}|Y=i}[T_{ij}(X)\ell(f(X),j)]
$$
$$
= \sum_{j\in[K]}\sum_{i\in[K]}\mathbb{P}(Y=i)T_{ij}\mathbb{E}_{\mathcal{D}|Y=i}[\ell(f(X),j)] + \sum_{j\in[K]}\sum_{i\in[K]}\mathbb{P}(Y=i)\text{Cov}_{\mathcal{D}|Y=i}(T_{ij}(X),\ell(f(X),j)).
$$

The first term could be transformed as:

$$
\sum_{j\in[K]}\sum_{i\in[K]}\mathbb{P}(Y=i)T_{ij}\mathbb{E}_{\mathcal{D}|Y=i}[\ell(f(X),j)]
$$
$$
= \sum_{j\in[K]}\left[T_{jj}\mathbb{P}(Y=j)\mathbb{E}_{\mathcal{D}|Y=j}[\ell(f(X),j)] + \sum_{i\in[K],i\neq j}T_{ij}\mathbb{P}(Y=i)\mathbb{E}_{\mathcal{D}|Y=i}[\ell(f(X),j)]\right]
$$
$$
= \underline{T}\mathbb{E}_{\mathcal{D}}[\ell(f(X),Y)] + \bar{\Delta}\mathbb{E}_{\mathcal{D}_\Delta}[\ell(f(X),Y)] + \sum_{j\in[K]}\sum_{i\in[K],i\neq j}T_{ij}\mathbb{P}(Y=i)\mathbb{E}_{\mathcal{D}|Y=i}[\ell(f(X),j)],
$$

where

$$
\underline{T} := \min_{j\in[K]} T_{jj}, \quad \bar{\Delta} := \sum_{j\in[K]}\Delta_j\mathbb{P}(Y=j), \quad \Delta_j := T_{jj} - \underline{T},
$$

and

$$
\mathbb{E}_{\mathcal{D}_\Delta}[\ell(f(X),Y)] := \begin{cases} \sum_{j\in[K]}\frac{\Delta_j\mathbb{P}(Y=j)}{\bar{\Delta}}\mathbb{E}_{\mathcal{D}|Y=j}[\ell(f(X),j)], & \text{if } \bar{\Delta} > 0, \\ 0 \text{ if } \bar{\Delta} = 0. \end{cases}
$$

Then

$$\mathbb{E}_{\widetilde{\mathcal{D}}}[\ell(f(X), \widetilde{Y})]$$
$$=\underline{T}\mathbb{E}_{\mathcal{D}}[\ell(f(X), Y)] + \bar{\Delta}\mathbb{E}_{\mathcal{D}_\Delta}[\ell(f(X), Y)] + \sum_{j\in[K]}\sum_{i\in[K], i\neq j}T_{ij}\mathbb{P}(Y=i)\mathbb{E}_{\mathcal{D}|Y=i}[\ell(f(X), j)],$$
$$+ \sum_{j\in[K]}\sum_{i\in[K]}\mathbb{P}(Y=i)\text{Cov}_{\mathcal{D}|Y=i}(T_{ij}(X), \ell(f(X), j))$$
$$=\underline{T}\mathbb{E}_{\mathcal{D}}[\ell(f(X), Y)] + \bar{\Delta}\mathbb{E}_{\mathcal{D}_\Delta}[\ell(f(X), Y)] + \sum_{j\in[K]}\sum_{i\in[K], i\neq j}T_{ij}\mathbb{P}(Y=i)\mathbb{E}_{\mathcal{D}|Y=i}[\ell(f(X), j)],$$
$$+ \sum_{j\in[K]}\sum_{i\in[K], i\neq j}\mathbb{P}(Y=i)\mathbb{E}_{\mathcal{D}|Y=i}[(T_{ij}(X) - T_{ij})(\ell(f(X), j) - \mathbb{E}_{\mathcal{D}|Y=i}[\ell(f(X), j)])]$$
$$+ \sum_{j\in[K]}\mathbb{P}(Y=j)\mathbb{E}_{\mathcal{D}|Y=j}[(T_{jj}(X) - T_{jj})(\ell(f(X), j) - \mathbb{E}_{\mathcal{D}|Y=j}[\ell(f(X), j)])]$$
$$=\underline{T}\mathbb{E}_{\mathcal{D}}[\ell(f(X), Y)] + \bar{\Delta}\mathbb{E}_{\mathcal{D}_\Delta}[\ell(f(X), Y)]$$
$$+ \sum_{j\in[K]}\sum_{i\in[K], i\neq j}\mathbb{P}(Y=i)\mathbb{E}_{\mathcal{D}|Y=i}[(T_{ij}(X) - T_{ij})(\ell(f(X), j) - \mathbb{E}_{\mathcal{D}|Y=i}[\ell(f(X), j)]) + T_{ij}\ell(f(X), j)]$$
$$+ \sum_{j\in[K]}\mathbb{P}(Y=j)\mathbb{E}_{\mathcal{D}|Y=j}[(T_{jj}(X) - T_{jj})(\ell(f(X), j) - \mathbb{E}_{\mathcal{D}|Y=j}[\ell(f(X), j)])]$$
$$=\underline{T}\mathbb{E}_{\mathcal{D}}[\ell(f(X), Y)] + \bar{\Delta}\mathbb{E}_{\mathcal{D}_\Delta}[\ell(f(X), Y)] + \sum_{j\in[K]}\sum_{i\in[K], i\neq j}\mathbb{P}(Y=i)\mathbb{E}_{\mathcal{D}|Y=i}[T_{ij}(X)\ell(f(X), j)]$$
$$+ \sum_{j\in[K]}\mathbb{P}(Y=j)\mathbb{E}_{\mathcal{D}|Y=j}[(T_{jj}(X) - T_{jj})\ell(f(X), j)]$$
$$=\underline{T}\mathbb{E}_{\mathcal{D}}[\ell(f(X), Y)] + \bar{\Delta}\mathbb{E}_{\mathcal{D}_\Delta}[\ell(f(X), Y)] + \sum_{j\in[K]}\sum_{i\in[K]}\mathbb{P}(Y=i)\mathbb{E}_{\mathcal{D}|Y=i}[U_{ij}(X)\ell(f(X), j)],$$

where
$$U_{ij}(X) = T_{ij}(X), \forall i \neq j, \quad U_{jj}(X) = T_{jj}(X) - T_{jj}.$$

The expected form of $\ell_{\text{CR}}$ on noisy distribution $\widetilde{\mathcal{D}}$ can be written as

$$\mathbb{E}_{\widetilde{\mathcal{D}}}[\ell_{\text{CR}}(f(x_i))] = -\beta\mathbb{E}_{\widetilde{\mathcal{D}}}\left[\mathbb{E}_{\mathcal{D}_{\widetilde{Y}|\widetilde{D}}}[\ell(f(x_i), \widetilde{Y})]\right]$$
$$= -\beta\int_{\widetilde{D}}\left[\mathbb{P}(\widetilde{D})\mathbb{E}_{\mathcal{D}_{\widetilde{Y}|\widetilde{D}}}[\ell(f(x_i), \widetilde{Y})]\right]$$
$$= -\beta\sum_{j\in[K]}\mathbb{P}(\widetilde{Y}=j)\mathbb{E}_{\mathcal{D}_X}[\ell(f(x_i), j)]$$
$$= -\sum_{j\in[K]}\sum_{i\in[K]}\mathbb{P}(Y=i)\mathbb{E}_{\mathcal{D}|Y=i}[\beta\mathbb{P}(\widetilde{Y}=j)\ell(f(x_i), j)].$$

Thus the expected form of the new regularized loss is

$$\mathbb{E}_{\widetilde{\mathcal{D}}}\left[\ell(f(X), \widetilde{Y}) + \ell_{\text{CR}}(f(x_i))\right] = \underline{T}\mathbb{E}_{\mathcal{D}}[\ell(f(X), Y)] + \bar{\Delta}\mathbb{E}_{\mathcal{D}_\Delta}[\ell(f(X), Y)]$$
$$+ \sum_{j\in[K]}\sum_{i\in[K]}\mathbb{P}(Y=i)\mathbb{E}_{\mathcal{D}|Y=i}[(U_{ij}(X) - \beta\mathbb{P}(\widetilde{Y}=j))\ell(f(X), j)]. \tag{8}$$

$\square$

## B.2 PROOF FOR THEOREM 1

**Theorem 1.** *For $\ell_{CA}(\cdot)$, solutions satisfying $f_{x_n}[i] > 0, \forall i \in [K]$ are not locally optimal at $(x_n, \tilde{y}_n)$.*

*Proof.* Let $\ell(\cdot)$ be the CE loss. Note this proof does not rely on whether the data distribution is clean or not. We use $\mathcal{D}$ to denote any data distribution and $D$ to denote the corresponding dataset. This

notation applies only to this proof. For any data distribution $\mathcal{D}$, we have

$$\mathbb{E}_{\mathcal{D}}\left[\ell(f(X), Y) - \mathbb{E}_{\mathcal{D}_{Y|D}}[\ell(f(x_n), Y)]\right]$$
$$=\mathbb{E}_{\mathcal{D}}[\ell(f(X), Y)] - \mathbb{E}_{\mathcal{D}_Y}[\mathbb{E}_{\mathcal{D}_X}[\ell(f(X), Y)]]$$
$$= -\int_{\mathcal{D}_X} dx \sum_{y \in [K]} \mathbb{P}(x, y) \ln f_x[y] + \int_{\mathcal{D}_X} dx \sum_{y \in [K]} \mathbb{P}(x)\mathbb{P}(y) \ln f_x[y]$$
$$= -\int_{\mathcal{D}_X} dx \sum_{y \in [K]} \ln f_x[y][\mathbb{P}(x, y) - \mathbb{P}(x)\mathbb{P}(y)].$$

The dynamical analyses are based on the following three assumptions:

A1. The model capacity is infinite (i.e., it can realize arbitrary variation).

A2. The model is updated using the gradient descent algorithm (i.e. updates follow the direction of decreasing $\mathbb{E}_{\mathcal{D}}[\ell(f(X), Y)] - \mathbb{E}_{\mathcal{D}_Y}[\mathbb{E}_{\mathcal{D}_X}[\ell(f(X), Y)]]$).

A3. The derivative of network function $\frac{\partial f(x;w)}{\partial w_i}$ is smooth (i.e. the network function has no singular point), where $w_i$'s are model parameters.

Denote the variations of $f_x[y]$ during one gradient descent update by $\Delta_y(x)$. From Lemma 1, it can be explicitly written as

$$\Delta_y(x) = f_x[y] \cdot \eta \int_{\mathcal{D}_X} dx' \sum_{y' \in [K]} [\mathbb{P}(x', y') - \mathbb{P}(x')\mathbb{P}(y')] \sum_{i \in [K]} G_i(x, y)G_i(x', y'), \quad (9)$$

where $\eta$ is the learning rate,

$$G_i(x, y) = -\frac{\partial g_y(x)}{\partial w_i} + \sum_{y' \in [K]} f_x[y']\frac{\partial g_{y'}(x)}{\partial w_i},$$

and $g_y(x)$ is the network output before the softmax activation. i.e.

$$f_x[y] = \frac{\exp(g_y(x))}{\sum_{y' \in [K]} \exp(g_{y'}(x))}.$$

With $\Delta_y(x)$, the variation of the regularized loss is

$$\Delta\mathbb{E}_{\mathcal{D}}[\ell(f(X), Y) + \ell_{\text{CR}}] = -\int_{\mathcal{D}_X} dx\, \mathbb{P}(x) \sum_{y \in [K]} \Delta_y(x)\frac{\mathbb{P}(y|x) - \mathbb{P}(y)}{f_x[y]}. \quad (10)$$

If the training reaches a steady state (a.k.a. local optimum), we have $\Delta\mathbb{E}_{\mathcal{D}}[\ell(f(X), Y) + \ell_{\text{CR}}] = 0$. To check the property of this variation, consider the following example. For a particular $x_0$, define

$$F(x_0) := \sum_{y \in [K]} \Delta_y(x_0)\frac{\mathbb{P}(y|x_0) - \mathbb{P}(y)}{f_{x_0}[y]}.$$

Split the labels $y$ into the following two sets (without loss of generality, we ignore the $\mathbb{P}(y|x_0) - \mathbb{P}(y) = 0$ cases):

$$\mathcal{Y}_{x_0;-} = \{y : \mathbb{P}(y|x_0) - \mathbb{P}(y) < 0\}$$

and

$$\mathcal{Y}_{x_0;+} = \{y : \mathbb{P}(y|x_0) - \mathbb{P}(y) > 0\}.$$

By assigning $\Delta_y(x_0) = a_y < 0, \forall y \in \mathcal{Y}_{x_0;-}$ and $\Delta_y(x_0) = b_y > 0, \forall y \in \mathcal{Y}_{x_0;+}$, one finds $F(x_0) > 0$ since $f_{x_0}[y] > 0$. Note we have an extra constraint $\sum_y \Delta_y(x_0) = 0$ to ensure $\sum_{y \in [K]} f_{x_0}[y] = 1$ after update. It is easy to check our assigned $a_y$ and $b_y$ could maintain this constraint by introducing a weight $N_{ab}$ to scale $b'_y$ as follows.

$$\sum_{y \in \mathcal{Y}_-} a_y + N_{ab} \sum_{y \in \mathcal{Y}_+} b'_y = 0, \ b_y = N_{ab}b'_y.$$

Let $B_\epsilon(x_0)$ be a $\epsilon$-neighbourhood of $x_0$. Since $f_x[y]$ is continuous, we can set $\Delta_y(x) = \frac{1}{2}(1 + \cos\frac{\pi\|x-x_0\|}{\epsilon})\Delta_y(x_0), \forall x \in B_\epsilon(x_0)$ and 0 otherwise. The coefficient $\frac{1}{2}(1 + \cos\frac{\pi\|x-x_0\|}{\epsilon})$ is added so that the continuity of $f_x[y]$ preserves. This choice will lead to $\Delta\widetilde{\mathbb{E}}_\mathcal{D}[\ell(f(X),Y) + \ell_{\mathrm{CR}}] < 0$. Therefore, for any $\ell_{\mathrm{CA}}(f(x_n),y_n)$ with solution $f_{x_n}[i] > 0, \forall i \in [K]$, we can always find a decreasing direction, indicating the solution is not (steady) locally optimal. Note $\mathcal{D}$ can be any distribution in this proof. Thus the result holds for the noisy distribution $\widetilde{\mathcal{D}}$. □

**Lemma 1.**

$$\Delta_y(x) = f_x[y] \cdot \eta \int_{\mathcal{D}_X} dx' \sum_{y' \in [K]} [\mathbb{P}(x',y') - \mathbb{P}(x')\mathbb{P}(y')] \sum_{i \in [K]} G_i(x,y)G_i(x',y').$$

*Proof.* We need to take into account the actual form of activation function, i.e., the softmax function, as well as the SGD algorithm to demonstrate the correctness of this lemma. The variation $\Delta_{y_0}(x_0)$ is caused by the change in network parameters $\{w_i\}$, i.e.,

$$\Delta_{y_0}(x_0) = \sum_{i \in [K]} \frac{\partial f_{x_0}[y_0]}{\partial w_i} \delta w_i, \tag{11}$$

where $\delta w_i$ are determined by the SGD algorithm

$$\delta w_i = -\eta \frac{\partial \mathbb{E}_\mathcal{D}[\ell(f(X),Y) + \ell_{\mathrm{CR}}]}{\partial w_i}$$

$$= \eta \sum_{x,y} \frac{\mathbb{P}(x,y) - \mathbb{P}(x)\mathbb{P}(y)}{f_x[y]} \frac{\partial f_x[y]}{\partial w_i}.$$

Plugging back to (11) yields

$$\Delta_{y_0}(x_0) = \eta \sum_{x,y} \frac{\mathbb{P}(x,y) - \mathbb{P}(x)\mathbb{P}(y)}{f_x[y]} \sum_{i \in [K]} \frac{\partial f_{x_0}[y_0]}{\partial w_i} \frac{\partial f_x[y]}{\partial w_i}.$$

To proceed, we need to expand $\frac{\partial f_x[y]}{\partial w_i}$. Taking into account the activation function, one has

$$f_x[y] = \frac{\exp(g_y(x))}{\sum_{y' \in [K]} \exp(g_{y'}(x))},$$

where $g_y(x)$ refers to the network output before passed to the activation function. Recall that, by our assumption, derivatives $\frac{\partial f(x;w)}{\partial w_i}$ are not singular. Now we have

$$\frac{\partial f_x[y]}{\partial w_i} = \frac{\partial e^{-g_y(x)}}{\partial w_i} \frac{1}{\sum_{y' \in [K]} e^{-g_{y'}(x)}} + e^{-g_y(x)} \frac{\partial}{\partial w_i}\left(\frac{1}{\sum_{y' \in [K]} e^{-g_{y'}(x)}}\right)$$

$$= \frac{-e^{-g_y(x)}}{\sum_{y' \in [K]} e^{-g_{y'}(x)}} \frac{\partial g_y(x)}{\partial w_i} + \frac{e^{-g_y(x)}}{\left(\sum_{y'' \in [K]} e^{-g_{y''}(x)}\right)^2} \sum_{y' \in [K]} e^{-g_{y'}(x)} \frac{\partial g_{y'}(x)}{\partial w_i}$$

$$= f_x[y]\left[-\frac{\partial g_y(x)}{\partial w_i} + \sum_{y' \in [K]} f_x[y'] \frac{\partial g_{y'}(x)}{\partial w_i}\right].$$

For simplicity, we can rewrite the above result as

$$\frac{\partial f_x[y]}{\partial w_i} = f_x[y]G_i(x,y),$$

where

$$G_i(x,y) := -\frac{\partial g_y(x)}{\partial w_i} + \sum_{y'} f_x[y'] \frac{\partial g_{y'}(x)}{\partial w_i}$$

is a smooth function.

Combining all the above gives $\Delta_{y_0}(x_0)$ as follows.

$$\Delta_{y_0}(x_0) = f_{x_0}[y_0] \cdot \eta \sum_{x,y} [\mathbb{P}(x,y) - \mathbb{P}(x)\mathbb{P}(y)] \sum_i G_i(x_0,y_0)G_i(x,y)$$

□

### B.3    PROOF FOR THEOREM 2

**Theorem 2.** *The sample sieve defined in (4) ensures that clean examples $(x_n, \tilde{y}_n = y_n)$ will not be identified as being corrupted if the model $f^{(t)}$'s prediction on $x_n$ is better than random guess.*

*Proof.* Let $y_n$ be the true label corresponding to feature $x_n$. For a clean sample, we have $\tilde{y}_n = y_n$. Consider an arbitrary DNN model $f$. With the CE loss, we have $\ell(f(x_n), y_n) = -\ln(f_{x_n}[y_n])$. According to Equation (4) in the paper, the necessary and sufficient condition of $v_n > 0$ is

$$\ell(f(x_n), \tilde{y}_n) + \ell_{\text{CR}}(f(x_n)) < \alpha_n \Leftrightarrow -\ln(f_{x_n}[y_n]) < -\frac{1}{K} \sum_{y \in [K]} \ln(f_{x_n}[y])$$

$$\Leftrightarrow -\ln(f_{x_n}[y_n]) < -\frac{1}{K-1} \sum_{y \in [K], y \neq y_n} \ln(f_{x_n}[y]).$$

By Jensen's inequality we have

$$-\ln\left(\frac{1 - f_{x_n}[y_n]}{K-1}\right) = -\ln\left(\frac{\sum_{y \in [K], y \neq y_n} f_{x_n}[y]}{K-1}\right) \leq -\frac{1}{K-1} \sum_{y \in [K], y \neq y_n} \ln(f_{x_n}[y]).$$

Therefore, when (sufficient condition)

$$-\ln(f_{x_n}[y_n]) < -\ln\left(\frac{1 - f_{x_n}[y_n]}{K-1}\right) \Leftrightarrow f_{x_n}[y_n] > \frac{1}{K},$$

we have $v_n > 0$. Inequality $f_{x_n}[y_n] > \frac{1}{K}$ indicates the model prediction is better than random guess. $\square$

### B.4    PROOF FOR THEOREM 4

Before proving Theorem 4, we need to show the effect of adding Term-2 to Term-1 in (5). Let $\epsilon_X < 0.5$ be the measure of separation among classes w.r.t feature $X$ in distribution $\mathcal{D}$, i.e., $\mathbb{P}(Y = Y^*|X) = 1 - \epsilon_X, (X, Y) \sim \mathcal{D}$, where $Y^* := \arg\max_{i \in [K]} \mathbb{P}(Y = i|X)$ is the Bayes optimal label. Let $\mathcal{D}'$ be the shifted distribution by adding Term-2 to Term-1 and $Y'$ be the shifted label. Then $\mathbb{P}(X|Y) = \mathbb{P}(X|Y'), \forall (X, Y) \sim \mathcal{D}, (X, Y') \sim \mathcal{D}'$ but $\mathbb{P}(Y')$ may be different from $\mathbb{P}(Y)$. Lemma 2 shows the invariant property of this label shift.

**Lemma 2.** *Label shift does not change the Bayes optimal label of feature $X$ when $\epsilon_X < \min_{\forall i, j \in [K]} \left(\frac{T_{jj}}{T_{ii} + T_{jj}}\right)$.*

*Proof.* Consider the shifted distribution $\mathcal{D}'$. Let

$$\underline{T}\mathbb{E}_{\mathcal{D}}[\ell(f(X), Y)] + \bar{\Delta}\mathbb{E}_{\mathcal{D}_\Delta}[\ell(f(X), Y)] = C\mathbb{E}_{\mathcal{D}'}[\ell(f(X), Y)],$$

where

$$\mathbb{E}_{\mathcal{D}'}[\ell(f(X), Y)] := \sum_{j \in [K]} \mathbb{P}(Y' = j)\mathbb{E}_{\mathcal{D}'|Y'=j}[\ell(f(X), j)],$$

and

$$\mathbb{P}(Y' = j) := \frac{T_{jj}\mathbb{P}(Y = j)}{C},$$

where $C := \sum_{j \in [K]} T_{jj}\mathbb{P}(Y = j)$ is a constant for normalization. For each possible $Y = i$, we have $\mathbb{P}(Y = i|X) \in [0, \epsilon_X] \cup \{1 - \epsilon_X\}, \epsilon_X < 0.5$. Thus

$$\mathbb{P}(X|Y = i) = \frac{\mathbb{P}(Y = i|X)\mathbb{P}(X)}{\mathbb{P}(Y = i)} \in [0, \frac{\epsilon_X \mathbb{P}(X)}{\mathbb{P}(Y = i)}] \cup \{\frac{\mathbb{P}(X)(1 - \epsilon_X)}{\mathbb{P}(Y = i)}\}.$$

Compare $\mathcal{D}'$ and $\mathcal{D}$, we know there is a label shift (Alexandari et al., 2020; Storkey, 2009), where $\mathbb{P}(X|Y=i) = \mathbb{P}(X|Y'=i)$ but $\mathbb{P}(Y)$ and $\mathbb{P}(Y')$ may be different. To ensure the label shift does not change the Bayes optimal label, we need

$$Y^* = \arg\max_{i\in[K]} \mathbb{P}(Y'=i|X) = \arg\max_{i\in[K]} \frac{\mathbb{P}(X|Y'=i)\mathbb{P}(Y'=i)}{\mathbb{P}(X)}, \ (X,Y') \sim \mathcal{D}.$$

One sufficient condition is

$$\frac{\epsilon_X \mathbb{P}(Y'=i)}{\mathbb{P}(Y=i)} < \frac{(1-\epsilon_X)\mathbb{P}(Y'=j)}{\mathbb{P}(Y=j)} \Rightarrow \epsilon_X < \min_{\forall i,j\in[K]} \left( \frac{T_{jj}}{T_{ii}+T_{jj}} \right)$$

$\square$

With Lemma 2, Assumption 1, and Assumption 2, we present the proof for Theorem 4 as follows.

**Theorem 4.** *(Robustness of the Confidence Regularized CE Loss) With Assumption 1 and 2, when*

$$\max_{i,j\in[K],X\sim\mathcal{D}_X} \frac{U_{ij}(X)}{\mathbb{P}(\widetilde{Y}=j)} \leq \beta \leq \min_{\mathbb{P}(\widetilde{Y}=i)>\mathbb{P}(\widetilde{Y}=j),X\sim\mathcal{D}_X} \frac{T_{ii}(X) - T_{ij}(X)}{\mathbb{P}(\widetilde{Y}=i) - \mathbb{P}(\widetilde{Y}=j)},$$

*minimizing* $\mathbb{E}_{\widetilde{\mathcal{D}}}[\ell(f(X),\widetilde{Y}) + \ell_{CR}(f(X))]$ *is equivalent to minimizing* $\mathbb{E}_{\mathcal{D}}[\ell(f(X),Y)]$.

*Proof.* It is easy to check $\epsilon_X = 0, \forall X \sim \mathcal{D}_X$ when Assumption 1 holds. Thus adding Term-2 to Term-1 in (5) does not change the Bayes optimal label. With Assumption 1, the Bayes optimal classifier on the clean distribution should satisfy $f^*(X)[Y] = 1, \forall(X,Y) \sim \mathcal{D}$. On one hand, when $\beta \geq \max_{i,j\in[K],X\sim\mathcal{D}_X} U_{ij}(X)/\mathbb{P}(\widetilde{Y}=j)$, we have

$$\beta_{ij}(X) := U_{ij}(X) - \beta\mathbb{P}(\widetilde{Y}=j) \leq 0, \forall i,j \in [K], X \sim \mathcal{D}_X.$$

In this case, minimizing the regularization term results in confident predictions. On the other hand, to make it unbiased to clean results, $\beta$ could not be arbitrarily large. We need to find the upper bound on $\beta$ such that $f^*$ also minimizes the loss defined in the latter regularization term. Assume there is no loss on confident true predictions and there is one miss-prediction on example $(x_n, y_n = j_1)$, i.e., the prediction changes from the Bayes optimal prediction $f_{x_n}[j_1] = 1$ to $f_{x_n}[j_2] = 1, j_2 \neq j_1$. Compared to the optimal one, the first two terms in the right side of (5) is increased by $T_{j_1,j_1}\ell_0$, where $\ell_0 > 0$ is the regret of one confident wrong prediction. Accordingly, the last term in the right side of (5) is increased by $(\beta_{j_1,j_1}(X) - \beta_{j_1,j_2}(X))\ell_0$. It is supposed that

$$T_{j_1,j_1}\ell_0 + (\beta_{j_1,j_1}(x_n) - \beta_{j_1,j_2}(x_n))\ell_0 \geq 0, \forall j_1,j_2 \in [K],$$

which is equivalent to

$$\beta(\mathbb{P}(\widetilde{Y}=j_1) - \mathbb{P}(\widetilde{Y}=j_2)) \leq T_{j_1,j_1}(x_n) - T_{j_1,j_2}(x_n), \forall j_1,j_2 \in [K].$$

Thus

$$\beta \leq \min_{\mathbb{P}(\widetilde{Y}=j_1)>\mathbb{P}(\widetilde{Y}=j_2),X\sim\mathcal{D}_X} \frac{T_{j_1,j_1}(X) - T_{j_1,j_2}(X)}{\mathbb{P}(\widetilde{Y}=j_1) - \mathbb{P}(\widetilde{Y}=j_2)}.$$

By mathematical inductions, it can be generalized to the case with multiple miss-predictions in the CE term. $\square$

## C OTHER JUSTIFICATIONS

In this section, we first compare $\ell_{CR}$ and entropy regularization in Section C.1 and highlight our superiority with both theoretical and experimental evidence, then show an example for explaining the variances incurred by label noise in Section C.2, and provide the risk bound in Section C.3 for training with the sieved examples that satisfy Corollary 1.

Table 4: Comparing $\ell_{\text{CR}}$ with ER on CIFAR-10.

| Method | Symm | | | Asymm | | |
| --- | --- | --- | --- | --- | --- | --- |
| | 0.2 | 0.4 | 0.6 | 0.1 | 0.2 | 0.3 |
| Baseline | 86.98 | 81.88 | 74.14 | 90.69 | 88.59 | 86.14 |
| Baseline + ER | 87.61 | 83.84 | 80.55 | 91.36 | 89.61 | 87.47 |
| Baseline + $\ell_{\text{CR}}$ | **90.70** | **88.29** | **82.10** | **92.41** | **91.02** | **90.53** |

### C.1 COMPARING $\ell_{\text{CR}}$ WITH ENTROPY REGULARIZATION

For simplicity, we consider two-class classification problem. Suppose for a given feature $x$, the probability of $x$ belonging to class 1 is $p$. The entropy regularization (ER) can be written as:

$$R_{\text{ER}}(p) = -(p \ln p + (1-p) \ln(1-p)), \tag{12}$$

while our regularization term is written as:

$$R_{\text{CR}}(p) = \ln p + \ln(1-p). \tag{13}$$

We have the following proposition:

**Proposition 1.** *$\ell_{\text{CR}}$ regularizes models stronger than the entropy regularization in terms of gradients.*

*Proof.* First notice that both $R_{\text{ER}}$ and $R_{\text{CR}}$ are symmetric functions around $p = 0.5$. Thus we can only consider the situation where $0 < p < 0.5$. The gradients w.r.t $p$ are:

$$\frac{\partial R_{\text{ER}}(p)}{\partial p} = -(\ln p - \ln(1-p)) = \ln(\frac{1}{p} - 1),$$

and

$$\frac{\partial R_{\text{CR}}(p)}{\partial p} = \frac{1}{p} - \frac{1}{1-p}.$$

Now we compare the absolute value of two gradients. When $0 < p < 0.5$, it is easy to check

$$\frac{\partial R_{\text{ER}}(p)}{\partial p} = \ln(\frac{1}{p} - 1) < \frac{1}{p} - 2 < \frac{1}{p} - \frac{1}{1-p} = \frac{\partial R_{\text{CR}}(p)}{\partial p},$$

and both gradients are larger than 0. Therefore, $\ell_{\text{CR}}$ has larger gradients than the entropy regularization, i.e., $\ell_{\text{CR}}$ has stronger regularization ability than ER. $\qquad\square$

We can also draw a figure to show this phenomenon. Figure 4 shows the value of $R_{\text{CR}}$ and $R_{\text{ER}}$ with respect to $p$. We can see the gradient of our regularization is larger than entropy regularization, resulting in a more confident prediction. We also perform an experiment to further show the evidence. Table 4 records comparison results which show our regularization achieves higher accuracy compared to the entropy term.

### C.2 CALCULATING $\text{var}_{\mathcal{D}}(\ell(f_{\mathcal{D}}^*(X), Y))$ AND $\text{var}_{\widetilde{\mathcal{D}}}[\ell(f_{\mathcal{D}}^*(X), \widetilde{Y}) + \ell_{\text{CR}}(f_{\mathcal{D}}^*(X))]$

Consider optimal classifier $f_{\mathcal{D}}^* := \arg\min_f \mathbb{E}_{\mathcal{D}}[\ell(f(X), Y)]$. Let $\ell_{\max}$ be the upper bound of the $\ell(\cdot)$ loss, and $\ell_{\min}$ be the lower bound of the $\ell(\cdot)$ loss. Denote $\varepsilon$ by the over noise rate (ratio of corrupted examples in all examples).

For $\text{var}_{\mathcal{D}}(\ell(f_{\mathcal{D}}^*(X), Y))$, we know the loss $\ell(f_{\mathcal{D}}^*(x_n), y_n) = \ell_{\min}$ for each example. Thus the variance is $\text{var}_{\mathcal{D}}(\ell(f_{\mathcal{D}}^*(X), Y)) = 0$.

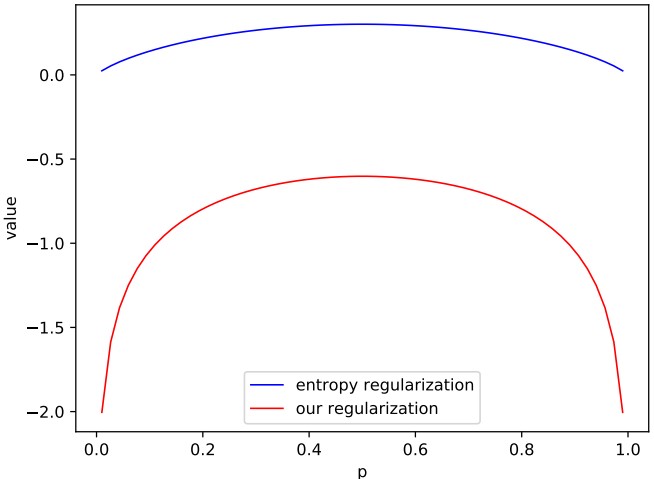

Figure 4: Comparing our regularization with entropy regularization .

For $\text{var}_{\widetilde{\mathcal{D}}}[\ell(f_{\mathcal{D}}^*(X), \widetilde{Y}) + \ell_{\text{CR}}(f_{\mathcal{D}}^*(X))]$, we know the loss $\ell(f_{\mathcal{D}}^*(x_n), \tilde{y} = y_n) = \ell_{\min}$, and the loss $\ell(f_{\mathcal{D}}^*(x_n), \tilde{y} \neq y_n) = \ell_{\max}$. Note

$$\ell_{\text{CR}} = \frac{(K-1)\ell_{\max} + \ell_{\min}}{K}$$

for each example. The expectation is

$$\mathbb{E}_{\widetilde{\mathcal{D}}}[\ell(f_{\mathcal{D}}^*(X), \widetilde{Y}) + \ell_{\text{CR}}(f_{\mathcal{D}}^*(X))] = \varepsilon\ell_{\max} + (1-\varepsilon)\ell_{\min} + \ell_{\text{CR}}.$$

Thus the variance is

$$\text{var}_{\widetilde{\mathcal{D}}}[\ell(f_{\mathcal{D}}^*(X), \widetilde{Y}) + \ell_{\text{CR}}(f_{\mathcal{D}}^*(X))]$$
$$=\varepsilon(\ell_{\max} + \ell_{\text{CR}} - (\varepsilon\ell_{\max} + (1-\varepsilon)\ell_{\min} + \ell_{\text{CR}}))^2 + (1-\varepsilon)(\ell_{\min} + \ell_{\text{CR}} - (\varepsilon\ell_{\max} + (1-\varepsilon)\ell_{\min} + \ell_{\text{CR}}))^2$$
$$=\varepsilon(1-\varepsilon)(\ell_{\max} - \ell_{\min})^2.$$

We know in this example,

$$\text{var}_{\widetilde{\mathcal{D}}}[\ell(f_{\mathcal{D}}^*(X), \widetilde{Y}) + \ell_{\text{CR}}(f_{\mathcal{D}}^*(X))] = \varepsilon(1-\varepsilon)(\ell_{\max} - \ell_{\min})^2 \gg \text{var}_{\mathcal{D}}(\ell(f_{\mathcal{D}}^*(X), Y)) = 0.$$

## C.3 ANALYSIS FOR THE RISK BOUND

Let $\widetilde{D}_{L^*}$ and $\widetilde{\mathcal{D}}_{L^*}$ be the set and the distribution of the sieved clean examples according to Corollary 1. We know they are supposed to contain only clean examples. Define $R_{\mathcal{D}}(f) := \mathbb{E}_{\mathcal{D}}[\ell(f(X), Y)]$, $f_{\mathcal{D}}^* := \arg\min_f R_{\mathcal{D}}(f)$, $\widehat{R}_{\widetilde{D}_{L^*},\gamma}(f) := \frac{1}{|L^*|}\sum_{n \in L^*}[\gamma(x_n)\ell(f(x_n), \tilde{y}_n)]$, $\hat{f}_{\widetilde{D}_{L^*},\gamma} := \arg\min_{f \in \mathcal{F}} \widehat{R}_{\widetilde{D}_{L^*},\gamma}(f)$, where $\gamma(X) := \mathbb{P}_{\mathcal{D}}(X)/\mathbb{P}_{\widetilde{\mathcal{D}}_{L^*}}(X)$ stands for the importance of each example to correct sample bias such that $R_{\mathcal{D}}(f) = \mathbb{E}_{\widetilde{\mathcal{D}}_{L^*}}[\gamma(X)\ell(f(X), \widetilde{Y})]$. The weight $\gamma(X)$ can be estimated by kernel mean matching (Huang et al., 2007) and its DNN adaption (Fang et al., 2020). Let $\widetilde{\mathcal{D}}_{L^*,X}$ be the marginal distribution of $\widetilde{\mathcal{D}}_{L^*}$ on $X$. For example, with a particular kernel $\Phi(X)$, the optimization problem is:

$$\min_{\gamma(X)} \quad \|\mathbb{E}_{\mathcal{D}_X}[\Phi(X)] - \mathbb{E}_{\widetilde{\mathcal{D}}_{L^*,X}}[\gamma(X)\Phi(X)]\|$$

$$\text{s.t.} \quad \gamma(X) > 0 \text{ and } \mathbb{E}_{\widetilde{\mathcal{D}}_{L^*,X}}[\gamma(X)] = 1.$$

Note the selection of kernel $\Phi(\cdot)$ is non-trivial, especially for complicated features. See (Fang et al., 2020) for a detailed DNN solutions.

Corollary 2 provides a risk bound for minimizing CE after sample sieve.

**Corollary 2.** *If $\gamma \cdot \ell$ is $[0, b]$-valued, then for any $\delta > 0$, with probability at least $1 - \delta$, we have*

$$R_{\mathcal{D}}(\hat{f}_{\widetilde{D}_{L^*}, \gamma}) - R_{\mathcal{D}}(f_{\mathcal{D}}^*) \leq 2\Re(\gamma \circ \ell \circ \mathcal{F}) + 2b\sqrt{\frac{\log(1/\delta)}{2|L^*|}},$$

*where the Rademacher complexity $\Re(\gamma \circ \ell \circ \mathcal{F}) := \mathbb{E}_{\widetilde{\mathcal{D}}_{L^*}, \boldsymbol{\sigma}}[\sup_{f \in \mathcal{F}} \frac{2}{|L^*|} \sum_{n \in L^*} \sigma_n \gamma(x_n) \ell(f(x_n), \tilde{y}_n)]$ and $\{\sigma_{n \in L^*}\}$ are independent Rademacher variables.*

*Proof.* The sieved clean examples may be biased due to the covariate shift caused by instance-based label noise. One solution to such shift is re-weighting $\widetilde{\mathcal{D}}_{L^*}$ to match $\mathcal{D}$ using importance re-weighting. Particularly, we need to estimate parameters $\gamma(X)$ such that

$$R_{\mathcal{D}}(f) = R_{\widetilde{\mathcal{D}}_{L^*}, \gamma}(f) := \mathbb{E}_{\widetilde{\mathcal{D}}_{L^*}}[\gamma(X)\ell(f(X), \widetilde{Y})].$$

With the optimal $\gamma(X)$, the ERM should be changed as

$$\hat{f}_{\widetilde{D}_{L^*}, \gamma} := \arg\min_{f \in \mathcal{F}} \widehat{R}_{\widetilde{D}_{L^*}, \gamma}(f),$$

where

$$\widehat{R}_{\widetilde{D}_{L^*}, \gamma}(f) := \frac{1}{|L^*|} \sum_{n \in L^*} [\gamma(x_n)\ell(f(x_n), \tilde{y}_n)].$$

Via Hoeffding's inequality, $\forall f$, w.p. at least $1 - \delta$, we have

$$|\widehat{R}_{\widetilde{D}_{L^*}, \gamma}(f) - R_{\widetilde{\mathcal{D}}_{L^*}, \gamma}(f)| \leq \Re(\ell \circ \mathcal{F}) + 2b\sqrt{\frac{\ln(1/\delta)}{2|L^*|}}.$$

Following the basic Rademacher bound (Bartlett & Mendelson, 2002) on the maximal deviation between the expected empirical risks:

$$\begin{aligned} &R_{\mathcal{D}}(\hat{f}_{\widetilde{D}_{L^*}, \gamma}) - R_{\mathcal{D}}(f_{\mathcal{D}}^*) \\ =&R_{\widetilde{\mathcal{D}}_{L^*}, \gamma}(\hat{f}_{\widetilde{D}_{L^*}, \gamma}) - R_{\widetilde{\mathcal{D}}_{L^*}, \gamma}(f_{\widetilde{\mathcal{D}}_{L^*}, \gamma}^*) \\ =&\left[\widehat{R}_{\widetilde{D}_{L^*}, \gamma}(\hat{f}_{\widetilde{D}_{L^*}, \gamma}) - \widehat{R}_{\widetilde{D}_{L^*}, \gamma}(f_{\widetilde{\mathcal{D}}_{L^*}, \gamma}^*) + \left(R_{\widetilde{\mathcal{D}}_{L^*}, \gamma}(\hat{f}_{\widetilde{D}_{L^*}, \gamma}) - \widehat{R}_{\widetilde{D}_{L^*}, \gamma}(\hat{f}_{\widetilde{D}_{L^*}, \gamma})\right) \right. \\ &\left. + \left(\widehat{R}_{\widetilde{D}_{L^*}, \gamma}(f_{\widetilde{\mathcal{D}}_{L^*}, \gamma}^*) - R_{\widetilde{\mathcal{D}}_{L^*}, \gamma}(f_{\widetilde{\mathcal{D}}_{L^*}, \gamma}^*)\right)\right] \\ \leq&0 + 2\max_{f \in \mathcal{F}} |\widehat{R}_{\widetilde{D}_{L^*}, \gamma}(f) - R_{\widetilde{\mathcal{D}}_{L^*}, \gamma}(f)| \\ \leq&2\Re(\gamma \circ \ell \circ \mathcal{F}) + 2b\sqrt{\frac{\ln(1/\delta)}{2|L^*|}}, \end{aligned}$$

where the Rademacher complexity $\Re(\gamma \circ \ell \circ \mathcal{F}) := \mathbb{E}_{\widetilde{\mathcal{D}}_{L^*}, \boldsymbol{\sigma}}[\sup_{f \in \mathcal{F}} \frac{2}{|L^*|} \sum_{n \in L^*} \sigma_n \gamma(x_n) \ell(f(x_n), \tilde{y}_n)]$ and $\{\sigma_{n \in L^*}\}$ are independent Rademacher variables. Therefore, we get Corollary 2.

$\square$

Corollary 2 informs us that, theoretically, the sample sieve is biased and $\gamma(X)$ is necessary to correct the selection bias. However, the error induced by estimating $\gamma(X)$ may degrade the performance. In addition, it is easy to check the optimal solution of performing direct ERM on the sieved clean examples is the same as $f_{\mathcal{D}}^*$ in expectation when Assumption 1 holds.

## D   MORE DETAILS AND RESULTS FOR EXPERIMENTS

We firstly show our training framework in Section D.1, then show implementation details and discussions in Section D.2. The algorithm for generating the instance-dependent label noise is provided in Section D.3. We show more experiments in Section D.4 and the ablation study in Section D.5.

## D.1 ILLUSTRATION OF THE TRAINING FRAMEWORK

Our experiments follows the framework shown in Figure 5.

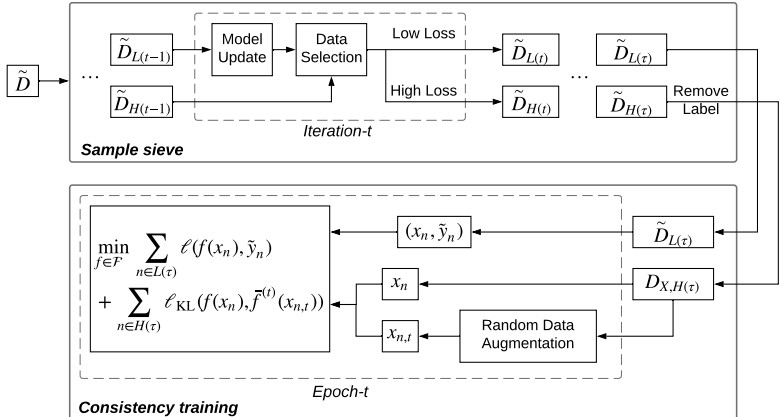

Figure 5: One example of CORES$^2$. $L(t)$: Indices of sieved clean examples. $H(t)$: Indices of sieved corrupted examples. $\widetilde{D}_{L(t)} := \{(x_n, \tilde{y}_n) : n \in L(t)\}$, $\widetilde{D}_{H(t)} := \{(x_n, \tilde{y}_n) : n \in H(t)\}$, $D_{X,H(\tau)} := \{x_n : n \in H(\tau)\}$.

## D.2 IMPLEMENTATION DETAILS AND MORE ANALYSIS

**Implementation details on CIFAR-10 and CIFAR-100 with instance-based label noise:** The basic hyper-parameters settings for CIFAR-10 and CIFAR-100 are listed as follows: mini-batch size (64), optimizer (SGD), initial learning rate (0.1), momentum (0.9), weight decay (0.0005), number of epochs (100) and learning rate decay (0.1 at 50 epochs). Standard data augmentation is applied to each dataset. CORES$^2$ and baseline share the same hyper-parameters setting except for $\alpha$ and $\beta$ in equation 2. When perform CORES$^2$, We first train network on the dataset for 10 warm-up epochs with only CE (Cross Entropy) loss. Then $\beta$ is linearly increased from 0 to 2 for next 30 epochs and kept as 2 for the rest of the epochs. The data selection is performed at the 30 epoch and $\alpha_{n,t}$ is set to $\frac{1}{K} \sum_{\tilde{y} \in [K]} \ell(\bar{f}^{(t)}(x_n), \tilde{y}) + \ell_{CR}(\bar{f}^{(t)}(x_n))$ in epoch-$t$ as the paper suggests.

When performing CORES$^{2\star}$, we used the sieved result at epoch-40. It is worth noting that at that time, the sample sieve may not reach the highest test accuracy. However, the division property brought by the confidence regularizer works well at that time. We use the default setting from UDA (Xie et al., 2019) to apply efficient data augmentation.

**Implementation details on Clothing-1M:** We train the network for 120 epochs on 1 million noisy training images. Batch-size is set to 32. The initial learning rate is set as 0.01 and reduced by a factor of 10 at 30, 60, 90 epochs. For each epoch, we sample 1000 mini-batches from the training data while ensuring the (noisy) labels are balanced. Mixup strategy is employed to further avoid the overfitting problem (Zhang et al., 2018; Li et al., 2020). $\beta$ is set to 0 at first 80 epochs, and linearly increased to 0.4 for next 20 epochs and kept as 0.4 for the rest of the epochs. It is worth noting that Clothing-1M actually does not satisfy our Assumption 2 since the class "Knitwear" (denoted by class-$i$) and the class "Sweater" (denoted by class-$j$) can not satisfy $T_{ii}(X) - T_{ij}(X) > T_{ii} - T_{jj}$. Note consistency training is not implemented on Clothing-1M.

**More analysis on $\beta$:** The value of $\beta$ mainly affects the sample sieve in CORES$^2$. From Theorem 3 and Theorem 4 in the paper, when $\beta$ is set to be small, we do not have the good division property. When $\beta$ is set to be large, the training is biased to the CE term. Figure 6 visualize this phenomenon. It can be seen that in the left and right figure, many clean examples and corrupted examples overlap together located in the left and right clusters, respectively.

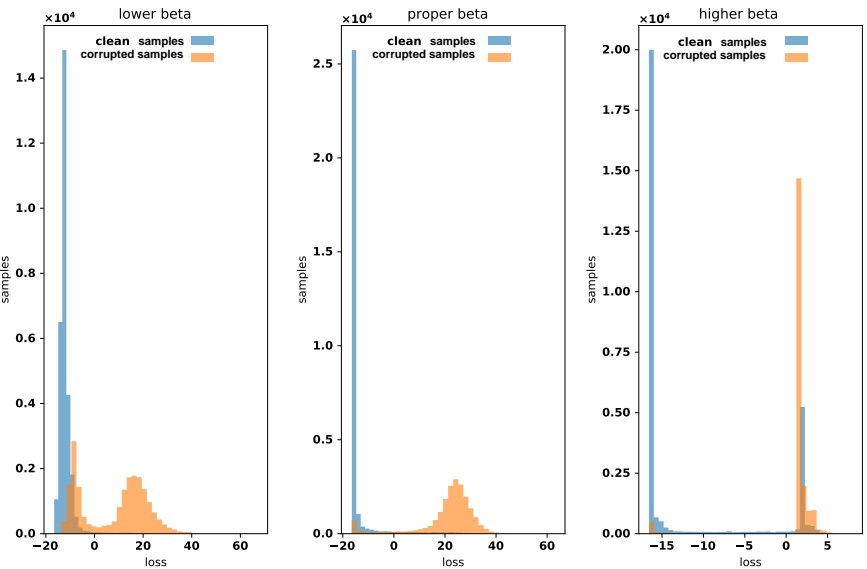

Figure 6: Analyzing how the value of $\beta$ influences the division. We set $\beta = 0.5, 2, 10$ for lower, proper, and higher beta settings, respectively.

---

**Algorithm 1** Instance-Dependent Label Noise Generation

---

**Input:**

    1: Clean examples $(x_n, y_n)_{n=1}^{N}$; Noise rate: $\varepsilon$; Size of feature: $1 \times S$; Number of classes: $K$.

**Iteration:**

    2: Sample instance flip rates $q_n$ from the truncated normal distribution $\mathcal{N}(\varepsilon, 0.1^2, [0, 1])$;

    3: Sample $W \in \mathcal{R}^{S \times K}$ from the standard normal distribution $\mathcal{N}(0, 1^2)$;

    **for** $n = 1$ to $N$ **do**

    4:     $p = x_n \cdot W$     // Generate instance dependent flip rates. The size of $p$ is $1 \times K$.

    5:     $p_{y_n} = -\infty$     // Only consider entries different from the true label

    6:     $p = q_n \cdot \mathrm{softmax}(p)$     // Let $q_n$ be the probability of getting a wrong label

    7:     $p_{y_n} = 1 - q_n$     // Keep clean w.p. $1 - q_n$

    8:    Randomly choose a label from the label space as noisy label $\tilde{y}_n$ according to $p$;

    **end for**

**Output:**

    9: Noisy examples $(x_i, \tilde{y}_n)_{n=1}^{N}$.

---

### D.3   GENERATING THE INSTANCE-DEPENDENT LABEL NOISE

In this section, we introduce how to generate instance-based label noise which is illustrated in Algorithm 1. Note this algorithm follows the state-of-the-art method (Xia et al., 2020). Define the noise rate (the global flipping rate) as $\varepsilon$. First, in order to control $\varepsilon$ but without constraining all of the instances to have a same flip rate, we sample their flip rates from a truncated normal distribution $\mathbf{N}(\varepsilon, 0.1^2, [0, 1])$, where $[0, 1]$ indicates the range of the truncated normal distribution. Second, we sample parameters $W$ from the standard normal distribution for generating instance-dependent label noise. The size of $W$ is $S \times K$, where $S$ denotes the length of each feature. For each instance $(x_n, y_n)$, we use Step 5 and Step 6 to ensure that the probability of getting a wrong label is $q_n$. Step 7 ensures the sum of all the entries of $p$ is 1.

Suppose there are two features: $x_i$ and $x_j$ where $x_i = x_j$. Then the possibility $p$ of these two features, calculated by $x \cdot W$, from the Algorithm 1, would be exactly the same. Thus the label noise is strongly instance-dependent.

Table 5: Comparison with the results reported by DivideMix (Li et al., 2020) on CIFAR-10. All methods use Pre-ResNet18 as the backbone. The last epoch test accuracy for each method is reported. The noise rate $\epsilon$ is defined as the probability of replacing the label with other labels including the true label.

| Dataset | Method | Symm | |
|---|---|---|---|
| | | 0.2 | 0.5 |
| | CE | 82.7 | 57.9 |
| | Bootstrap (Reed et al., 2014) | 82.9 | 58.4 |
| | Forward $T$ (Patrini et al., 2017) | 83.1 | 59.4 |
| | Co-teaching+ (Yu et al., 2019) | 88.2 | 84.1 |
| CIFAR-10 | Mixup (Zhang et al., 2018) | 92.3 | 77.6 |
| | P-correction (Yi & Wu, 2019) | 92.0 | 88.7 |
| | Meta-Learning (Li et al., 2019) | 92.0 | 88.8 |
| | M-correction (Arazo et al., 2019) | 93.8 | 91.9 |
| | DivideMix (Li et al., 2020) | 95.7 | 94.4 |
| | CORES$^{2\star}$ | **95.9** | **94.5** |

Table 6: The best epoch accuracy for each method on Tiny-ImageNet.

| Dataset | Model | Method | Symm | |
|---|---|---|---|---|
| | | | 0.2 | 0.5 |
| | | MAE (Ghosh et al., 2017) | 2.36 | 1.22 |
| Tiny-ImageNet | ResNet18 | GCE (Zhang & Sabuncu, 2018) | 69.84 | 66.31 |
| | | MentorNet (Jiang et al., 2017) | 59.12 | 53.83 |
| | | CORES$^{2\star}$ | 73.47 | 71.07 |

Table 7: Comparing CORES$^2$ (without consistency training) with other noise-robust methods on CIFAR-10.

| Method | Symm | | | Asymm | | |
|---|---|---|---|---|---|---|
| | 0.2 | 0.4 | 0.6 | 0.1 | 0.2 | 0.3 |
| Cross Entropy | 86.98 | 81.88 | 74.14 | 90.69 | 88.59 | 86.14 |
| Forward $T$ (Patrini et al., 2017) | 88.11 | 83.27 | 75.34 | 90.11 | 89.42 | 88.25 |
| Truncated $L_q$ (Zhang & Sabuncu, 2018) | 89.70 | 87.62 | 82.70 | 90.43 | 89.45 | 87.10 |
| $L_{\text{DMI}}$ (Xu et al., 2019) | 88.74 | 83.04 | 76.51 | 90.28 | 89.04 | 87.88 |
| CORES$^2$ (without consistency training) | 90.70 | 88.29 | 82.10 | 92.41 | 91.02 | 90.53 |

## D.4 MORE EXPERIMENTS ON CIFAR-10 AND TINY-IMAGENET

In this section, we compare CORES$^2$ with more methods on CIFAR-10 and Tiny-Imagenet. Table 5 records the comparison results with recent benchmark methods. Table 6 compares CORES$^2$ with other methods on Tiny-ImageNet. Both tables show that CORES$^2$ achieves competitive results.

## D.5 ABLATION STUDY

**CORES$^2$ (without consistency training)**: By optimizing loss in (2), the model can be forced to concentrate only on clean examples. Thus even without consistency training, the network trained by CORES$^2$ is also noise-robust. Table 7 compares CORES$^2$ with other noise-robust methods which do not apply semi-supervised setting in the framework. We can see CORES$^2$ still achieves the best performance among all the methods.

**CORES$^2$ without confidence regularization or dynamic data selection**: The loss in equation 2 consists of data selection strategy and confident regularization term. To see how they influence the final accuracy, we perform the ablation study to show their effect on Table 8. The first row of Table 8 corresponds to the traditional CE loss. The second row corresponds to the sample sieve with CE

Table 8: Analysis of each component of CORES$^2$ on CIFAR-10. All the methods use ResNet-34.

| Sample Sieve | | Consistency training | Symm | | | Asymm | | |
| Data selection | Regularization | | 0.2 | 0.4 | 0.6 | 0.1 | 0.2 | 0.3 |
|---|---|---|---|---|---|---|---|---|
| × | × | × | 86.67 | 81.44 | 74.63 | 90.18 | 88.43 | 87.27 |
| ✓ | × | × | 90.15 | 86.98 | 78.36 | 91.59 | 90.89 | 88.51 |
| ✓ | ✓ | × | 90.70 | 88.29 | 82.10 | 92.41 | 91.02 | 90.53 |
| ✓ | ✓ | ✓ | 95.73 | 93.76 | 89.78 | 96.05 | 95.18 | 94.67 |

loss. The third row is the typical CORES$^2$. The last row is CORES$^{2\star}$. We can see both the dynamic sample sieve in (4) and the confidence-regularized model update in (3) show positive effects on the final accuracy, which suggests the rationality of CORES$^2$.

