# OpenReview forum: "Learning with Instance-Dependent Label Noise: A Sample Sieve Approach"
_ICLR.cc/2021/Conference — ICLR 2021 Poster_

### Official Review · AnonReviewer2 · 2020-10-19
**Solid work! Nice contribution!**

**Rating:** 8
**Confidence:** 5

**Review:**

***quality***
This paper is quite well-written. The contribution is critical in instance-dependent label noise learning. Moreover, both the theoretical and empirical justifications are convincing.

***clarity***
Although this paper contains heavy mathematics, it is not difficult to understand. I can see that the authors have spent a lot of efforts in paper writing.

***originality***
In this paper, the authors proposed a novel sample sieve approach for instance-dependent label noise learning. The proposed model is novel, and the theoretical contributions are also new to the community.

***significance***
The proposed algorithm is simple, but the theory behind is rich. I like this kind of work, so I feel that the significance of this paper is high for future research.

***pros and cons***

Pros:
1. The topic is very important for realistic machine learning problems, and is helpful for reducing the human annotation efforts.
2. The theoretical study of this paper is quite impressive.
3. The experimental results show that the proposed method achieves SOTA performance.

Cons:
1. The authors claim that their method does not need to estimate the label transition probability or noise rate, which I think is nice! However, it would be better if the authors can explain why the proposed method can avoid this, namely which “component” helps to avoid the estimation for noise rate?
2. Since this work is an extension of cross-entropy loss to dealing with label noise, I think the comparison with “Symmetric Cross Entropy for Robust Learning with Noisy Labels”(ICCV 2019) is necessary, as this paper also aims to design a robust loss via modifying cross-entropy loss.
3. I feel that the sample sieve/filtering process (i.e. Eqs. 1-4) looks like self-paced learning (SPL) (see “Self-Paced Robust Learning for Leveraging Clean Labels in Noisy Data”, AAAI 20), as SPL also selects some “important” data for training in each iteration. Maybe the authors can discuss the relationship between these two methods?
4. The authors misuse the terms “sample” and “example”. Statistically, we say that we have a sample X={x_1,x_2,…,x_n} from some distribution, in which every x_i is an example.
5. Some recent typical works on label noise learning can be cited, such as “Are Anchor Points Really Indispensable in Label-Noise Learning?”(NeurIPS 19) and “A Bi-level Formulation for Label Noise Learning with Spectral Cluster Discovery” (IJCAI 20).

---

> ### Author Response · Authors · 2020-11-14
> **Intuitions for our approach and comparison to related works. Thanks for your recognition!**
>
> Thanks for your recognition of our paper. Your concerns are addressed as follows.
>
> **A1: More explanations:**
> In Theorem 3, we provide a generic machinery for anatomizing noisy datasets. From Eq. (5) in this theorem, for instance, we can see the effects caused by $U_{ij}(X)$ (capturing the information in the noise transition matrix $T(X)$) may be "canceled out" or "reversed" by $\beta \mathbb P(\widetilde Y=j)$ with a properly tuned $\beta$. Intuitively, with an appropriate $\beta$, all the effects of $U_{ij}(X), i\ne j$ can be reversed and we will get a negative loss punishing the classifier to predict class-$j$ when the clean label is $i$. We discussed similar intuitions before presenting Theorem 4 in the initial submission and polished it in the revised version.
>
> **A2: Related works:**
> Thanks for suggesting more related works. We have just experimented with SCE [R1] on CIFAR10 and CIFAR100 using instance-dependent noise, and obtained the following results:
> SCE CIFAR10~ Inst. 0.2: 89.11, Inst. 0.4: 72.04, Inst. 0.6: 44.83.
> SCE CIFAR100 Inst. 0.2: 59.87, Inst. 0.4: 41.76, Inst. 0.6: 23.41.
> We have added these results in Table 1 of the revised version.
> Most of the SPL-based methods, e.g. [R2], GCE (Zhang \& Sabuncu, 2018), Co-teaching (Han et al., 2018), Co-teaching+ (Yu et al., 2019), and JoCoR (Wei et al., 2020), rely on critical thresholds for determining whether the example is clean or corrupted, and the thresholds are often set/tuned empirically or manually. Our closed-form thresholds $\alpha_{n,t}$ can be directly calculated based on the model prediction and do not require extra estimation. It is interesting to consider further improving the dynamic sample sieve with the help of a small set of well-labeled examples with little data corruption as done in [R2]. We have cited all these related works [R1-R4] and addressed them properly in the revised version.
>
>
> Thanks for suggesting the proper use of "sample" and "example" -- we have fixed them in the revision.
>
> [R1] Wang, Yisen, et al. "Symmetric cross entropy for robust learning with noisy labels." CVPR 2019.
>
> [R2] Zhang, Xuchao, et al. "Self-Paced Robust Learning for Leveraging Clean Labels in Noisy Data." AAAI 2020.
>
> [R3] Xia, Xiaobo, et al. "Are Anchor Points Really Indispensable in Label-Noise Learning?." Advances in Neural Information Processing Systems. 2019.
>
> [R4] Luo, Yijing, Bo Han, and Chen Gong. "A Bi-level Formulation for Label Noise Learning with Spectral Cluster Discovery."

---

### Official Review · AnonReviewer4 · 2020-10-27
**Novel noise-robust loss for image classification**

**Rating:** 6
**Confidence:** 4

**Review:**

Summary

The paper introduces a noise-robust loss function CORES2, motivated by peer loss. The novel loss adds a regularization term that promotes confident prediction and pushes the model prediction away from the prior of the label. Using this loss function, the authors propose a dynamic sample sieve to separate the clean data and corrupted data on-the-fly, by the magnitude of CORES2 loss. The author's approach is to rule out samples whose losses are larger than an adaptive threshold. Importantly, the process of sieving successfully sieves out corrupted samples, both in a theory of 'better than random guess classifier' and in practice. The authors, then show that the proposed CORES2 can be decoupled under the instance-dependent noise setting. Then CORES2 is proved to be noise-robust, which means CORES2 is equivalent to minimizing the original cross-entropy loss. They also show a principle approach for finding the hyperparameters $\beta$. Further, a consistency loss is adopted after sample sieve on the corrupted samples. The author conducts extensive experiments, including CIFAR10, CIFAR100, and Clothing1M under different settings of noise. CORES2 achieves the SOTA results in all the experiments.

Contributions

i) Proposal of a novel confidence regularized loss for image classification and a novel algorithm that dynamically sieves out corrupted samples basing on their loss.

ii) The proposed loss is proved to be noise-robust. This theoretical result is valuable since less the instance-based noise is not well studied. A principle way to choose the hyperparameter $\beta$.

iii) Application to CIFAR10, CIFAR100, and Clothing1M (real-world dataset), with the extensive comparison.

Issues:

i) The motivation behind the dynamic sample sieve is unclear to me. One hypothesis is that if we set CORES2 as the objective, the model will fit clean samples much faster than corrupted samples at the beginning of training.  Hence the model can sieve the corrupted samples out. However, Fig2 shows that the cross-entropy can also separate clean/corrupted samples at the early training stage. I am curiously about the performance of cross-entropy loss + dynamic sample sieve. The comparison between CORES2 and cross-entropy in Fig2 is unfair.

Theoretically, the paper only shows that once the model is better than random guess, the dynamic sample sieve will not sieve clean samples out. And the assumption is $ f_{x_n}(y_n) > 1/K$. But $f_{x_n}(\tilde{y}_n) > 1/K$ would also frequently happen, for $y_n\not=\tilde{y}_n$, which means the dynamic sample sieve will also keep these corrupted samples.
Overall, I think the high-level insight into the sieving process is unclear in the paper.

ii) The training stability of the confidence regularizer: The authors show that confidence regularizer can promote confidence prediction. So some entries in $f(x)$ is near 0, and will make the optimization process unstable since the confidence regularizer -> inf in this case.
There is another concern related to theorem 4. Since the Bayes optimal classifier is $f^*_x[y]=1$, its CORES2 loss would be -inf, which seems problematic.

iii) Selection for $ \beta$, according to theorem 4: theorem 4 provides a principal way for estimation of \beta in different datasets. However, to estimate $ \beta$, we have to estimate the noise transition matrix beforehand. And there may exist some adversary samples that make the rough estimation impossible. In another perspective, we can also give some meaningful upper/lower bound for $\beta$ in the instance-independent noise setting. So it's unclear to me what makes the difference in the instance-dependent setting if we do not go through all the samples but do a rough estimation.

iv) One motivation of Confidence Regularizer is that confident prediction counters the overfitting of noise labels. But if the model capacity is sufficiently large, I think it can both overfit CORES2, and overfit those noise samples.

Minors:

a) The assumption shows that CORES2 does not favor the non-diagonal dominant noise rate, which means the diagonal entries in the noise transition matrix T can be smaller than some entries in the same row. Does CORES2 fail in this setting in practice?


Overall, the paper's approach is novel and easy to implement. I am willing to raise my score if the authors can address my issues.

#### EDIT------------

I think the authors replied to some of my concerns in a convincing way, hence I raise the score to 6.

Unfortunately, I think the theoretical analysis for the noise-robust loss is orthogonal to their sampling sieving approach. And the analysis for choosing $\beta$ does not dependent on their instance-dependent noise settings. We can get the same $\beta$ by very rough estimation(their approach) in instance-independent noise settings. In addition, I guess $f_x^*[y]=1$ is still problematic in the theoretical analysis since CORES2=\inf given the ideal classifier.

Overall, following author's response, I am leaning towards acceptance, but will let the AC judge the importance of the points above for the final decision.

---

> ### Author Response · Authors · 2020-11-14
> **Fair comparison to "CE + dynamic sample sieve"& high-level intuitions. Thanks for your comments and recognition! (part 1)**
>
> **A1: compare cross-entropy loss + dynamic sample sieve:**
>
> Thanks for your comment. The proposal "cross-entropy loss + dynamic sample sieve" is indeed an effective method under instance-independent label noise such as symmetric noise and asymmetric noise. There are pretty effective works adopting similar sample selection ideas (but our thresholds $\alpha_n$ for sieving are novel): MentorNet (Jiang et al., 2017), Co-teaching (Han et al., 2018), Co-teaching+ (Yu et al., 2019), etc. In this paper, we did compare our method with these approaches. For example, in Figure 3, both Co-teaching and CORES$^2$ work well in the symmetric noise setting, but performance degradation occurs in Co-teaching with $40\%$ and $60\%$ instance-dependent label noise.
>
> We agree with the reviewer that the comparison between CORES$^2$ and CE in Figure 2 is unfair. The main purpose of presenting Figure 2 is visualizing the "division" property of our sample sieve. As we can see, with symmetric (instance-independent) label noise, CE also provides good division thus the "CE + sample selection" approaches work well as shown in Figure 3, Table 1, and Table 5, 6 (in Appendix). Although the current comparison might look "unfair", no example is sieved out at epoch-20 since the sample sieve starts at epoch-30 in our experiments as mentioned in Appendix D.2. Thus the comparison at epoch-20 is "fair". We present the result of "CE + dynamic sample sieve" in Figure 2 of the revised version. From the revised Figure 2(b) and Figure 2(f), "CE+Sieve" is hard to perfectly sieve out corrupted examples in either symmetric noise or instance-dependent noise. A lot of corrupted examples with symmetric noise are sieved out by CE Sieve in Figure 2(b) while few corrupted examples with instance-dependent noise are sieved out in Figure 2(f), indicating the instance-dependent label noise is indeed more challenging. Note CE primarily helps learn a noisy distribution, while our regularized loss directs the training towards learning the underlying clean distribution, which is critical. We highlight this difference in Contribution-1 of the revised version. To be more accurate, the hypothesis in Issue-1 could be "if we set CORES2 as the objective, the model will fit (the underlying) clean examples during training". More explanations are available in A5.
>
> **A2: Motivation or high-level insights to the sieving process**
>
> The reviewer is absolutely correct that the dynamic sample sieve may include corrupted examples (as illustrated in Figure 1). We think there are generally two different philosophies toward sample selection: picking up clean examples or ruling out corrupted examples. Both methods have their own advantages. If we have one approach that is guaranteed to only select clean examples and does not address the quality of the "sieved out" examples, it will keep a smaller number of clean examples than that in the original noisy dataset and some clean examples may be sieved out. This approach may work well since the remaining examples are all clean, while we may lose the information captured by those sieved-out clean examples. It is hard to quantify the importance of the dropped information while this approach may not be robust enough in the challenging instance-dependent setting. On the other hand, our theory guarantees only corrupted examples are sieved out, indicating all the clean information is preserved. When some corrupted examples are safely dropped, the overall noise rate of the new dataset is guaranteed to decrease. Then with a dynamic sample sieve, the overall noise rate will be non-increasing and the performance will expect to progressively increase. This is the high-level design intuition of the sample sieve. Note it is hard to theoretically guarantee that the performance is always non-decreasing in the former case where we keep only a part of clean examples at the cost of dropping the other clean examples (some clean features may be completely dropped). In order to make the case "ruling out corrupted examples" optimal, we need to refer to the robustness of the regularized CE loss (Theorem 4). Theoretically, when Theorem 4 holds, the sample sieve can achieve the optimal performance (all the corrupted examples are sieved out) as claimed in Corollary 1. We highlight the difference in designing philosophies at the bottom of page 4.

---

> ### Author Response · Authors · 2020-11-14
> **responses to other concerns (part2)**
>
> **A3: CORES2 loss would be -inf**
> We do have this numerical issue, though happening rarely. $f_x(y) = 1$ is an ideal case in our theoretical analysis. In our code, we add $10^{-8}$ to all the probabilities after the soft-max layer to ensure numerical stability. We add a footnote at the beginning of Section 4 to clarify this in our revised version.
>
>
> **A4: Selection for $\beta$**
> Accurately estimating the noise transition matrix $T(X)$ in the instance-dependent label noise setting is challenging since there are about $N \cdot K^2$ parameters to be estimated. Recent work (Xia et al., 2020) assumes the instance-dependent label noise can be divided into several parts and assumes it to be instance-independent within each part and significantly reduces the number of parameters to be estimated. Therefore, we think it is not an ideal approach to firstly estimating or roughly estimate $T(X)$ and then calculating $\beta$. Fortunately, as shown in Theorem 4, there is a range for $\beta$ and therefore CORES$^2$ is more robust as long as $\beta$ is not too far off. The above result also implies that even if we leverage the estimation of the transition matrices to estimate $\beta$, our approach would be more robust w.r.t. the estimation errors. In Appendix D.1, we show the loss distribution with different $\beta$ settings. In our experiment, we do not estimate $T(X)$ since our method is relatively stable with $\beta$. Additionally, we do not fine-tune $\beta$ for different noise rates. For example, we fix $\beta = 2$ for all the CIFAR10 experiments.
>
>
>
> **A5: Overfitting with infinite model capacity**
> When the model capacity is sufficiently large and pure CE loss is adopted, the DNN will memorize all the examples no matter it is clean or corrupted. However, Theorem 4 shows minimizing the regularized CE loss on a noisy distribution is equivalent to minimizing the pure CE loss on the corresponding clean distribution in terms of the expectation. With this theory, we believe the DNN with CORES$^2$ will not overfit to label noise when the dataset is sufficiently large. In real experiments with a limited size of the dataset, as discussed in Section 3.3, assumptions in Theorem 4 cannot perfectly hold thus we need to train with the sieved examples to improve the performance.
>
>
> **A6: Non-diagonal elements**
> Thanks for your comments. We do need $T_{ii}(X)-T_{ij}(X)>0$ (typo revised in Assumption-2, sorry for the confusion). But CORES$^2$ does not fail in the scenario where $T_{ii}(X)-T_{ij}(X)<0$ for some $X$. In experiments, when the instance-independent noise rate is set to $0.6$, the case that $T_{ii}(X) - T_{ij}(X) < 0$ may occur since its expectation $T_{ii} = 1-0.6=0.4$ and $T_{ij}(X)$ may be greater than $T_{ii}(X)$ (e.g. $T_{ii}(X)=0.4$, $T_{ij}(X) = 0.6$) if the feature $X$ is easily to be identified as class-$j$. Thus Assumption-2 does not hold in this setting. From Table 1 and Table 2, when the overall noise rate $\varepsilon = 0.6$, we do observe performance degradation but CORES$^2$ still performs well.

---

### Official Review · AnonReviewer3 · 2020-10-28
**Review for "Learning with Instance-Dependent Label Noise: A Sample Sieve Approach"**

**Rating:** 6
**Confidence:** 4

**Review:**

The authors of the paper propose a new method, the CORES (COnfidence REgularized Sample Sieve), to tackle the important problem of learning under instance dependent label noise. The proposed method, in essence, involves the use of a confidence regularization term that encourages more confident predictions and a sieving process to remove the samples with large losses. Theoretical justification and empirical experiments were conducted to demonstrate the effectiveness of the proposed method.

All in all, the paper is clearly written and easy to follow. The proposed method seems technically sound and the motivation for the proposal is explained clearly. One major complaint I have for the paper is the lack of novelty of the paper. The two important building blocks of the paper, the confidence regularizer, and the sample sieve are derived from previous papers. Specifically, in my opinion, the confidence regularizer is a marginal extension of the "peer loss" [1], and the sample sieve algorithm is essentially the same as that proposed in [2], the only difference being a different choice of loss function for training and sieving, to the best of my knowledge and understanding. I think it is worth commenting on this very relevant line of work in Section 2.2. In addition, it would be interesting if the authors of the paper could offer some insights on why the proposed sieving strategy works better than the one previously proposed [2] based on softmax probability. All in all, with the lack of novelty addressed above, I think the submission is marginally below the acceptance threshold.

Other comments:
1. I find the intuitive justification for confidence regularization in Section 2.1 to be quite unconvincing. Specifically, it was stated that "when model overfits to the noise, its predictions often become less confident". From my understanding, this is not necessarily true at all. In fact, it was previously demonstrated that deep NNs can even perfectly overfit to datasets with randomly assigned labels? From this perspective, wouldn't encouraging confidence make the model overfit harder to the noisy labels? I would appreciate if the authors of the paper could provide further insights and intuition on why the introduced confidence regularization improves noise robustness.


[1] Yang Liu and Hongyi Guo. Peer loss functions: Learning from noisy labels without knowing noise rates. In Proceedings of the 37th International Conference on Machine Learning, ICML ’20, 2020.

[2] Zhilu Zhang and Mert Sabuncu. Generalized cross entropy loss for training deep neural networks with noisy labels. In Advances in neural information processing systems, pp. 8778–8788, 2018.

--------------------------------------------------------------------------------------------------------------
The authors of the paper addressed carefully the concerns I raised above. As such, I am raising my score to a 6, and would like to recommend accepting this paper.

---

> ### Author Response · Authors · 2020-11-14
> **Thanks for your comments & novelty of our approach (part1)**
>
> Thanks for your comments and suggestions. We would like to argue that our proposed method is indeed novel to the literature since we discover the novel confidence regularization property of CR by reforming peer loss, technically break through the bottlenecks of multi-class extensions and the instance-independent assumption in peer loss, and propose a novel sample sieve relying on a different philosophy (sieving out corrupted examples) from existing works. We summarize our novelty in **A1**, compare with peer loss and GCE in **A2** and **A3**, give intuitive explanations on why our sieve is better than GCE in **A4**, and discuss the overfitting of DNNs in **A5**.
>
> **A1: Novelty (in a nutshell)**
> We are happy to see that our technical contributions are appreciated. Although both the forms of loss and the sample sieve method may look ``similar'' to the existing literature, we do make critical modifications to both components. Comparing with peer loss, we would like to highlight our observed novel property of encouraging confident predictions by adding the proposed confidence regularizer (CR), and the advantage of CR in handling instance-dependent label noise. Note peer loss only guarantees the performance in binary classifications and particular multi-class extension when $T_{ij} = T_{kj}, \forall j\ne k\ne i$. However, in the challenging instance-dependent label noise, $T_{ij}(x)$ for each example $x$ could be different thus this assumption adopted in peer loss does not hold. The technical breakthrough of these bottlenecks of peer loss with our CR is novel and important. Comparing with GCE, we would like to highlight our design of a sample sieve following a totally different philosophy (sieving out corrupted examples), which enables the closed-form thresholds $\alpha_{n,k}$ along with the application of our confidence regularized loss. Moreover, we theoretically guarantee each adaptation and their combinations. Note the quality of a sample sieving procedure is unclear without the implementation of CR. Detailed technical contributions compared with peer loss and GCE are available in the following two responses. Summarizing above, we would argue that our efforts in reforming peer loss and developing a theoretically sound sieving process are non-trivial contributions. As also acknowledged by Reviewer-2, we believe such a simple but theoretically-guaranteed approach would contribute to the literature and can potentially inspire a line of future works.

---

> > ### Comment · AnonReviewer3 · 2020-11-17
> > **Thanks for such a detailed rebuttal!**
> >
> > I would like to thank the authors for such a detailed rebuttal, which helped me resolve a lot of the concerns about the paper. I have updated my score. below are some additional questions and comments:
> >
> > 1. Section A4 of your rebuttal was very helpful for me to understand why the proposed method can be better than previously proposed filtering strategies like GCE. It would be great if you could include such a detailed explanation and intuition on why $\alpha_{n,t}$ is better than an empirically determined threshold (I see that you added a comment that "Intuitively, the specially designed thresholds $\alpha_{n,t}$ for each example should be more accurate than a single threshold for the whole dataset". But why is this the case?). In addition, an additional look at your threshold  $\alpha_{n,t}$ and Equation 4 on page 4 made me slightly confused. Specifically, is $\ell_{CR}(\bar{f}^{(t)}(x))$ the same as $\ell_{CR}(f^{(t)}(x))$? If so, why do you have this additional term in the indicator function of equation 4?
> >
> > 2. You commented that in A5 that "Our proposal of using the CR term to avoid fitting noisy labels by encouraging confident predictions at the early stage.". While I understand that NNs tend to learn the clean samples first before overfitting to noisy labels based on the previous study, I wonder if the same observations would also be seen with the additional CR term during training? Intuitively, CR term would only be beneficial if the rate of increase in confidence during training is faster for clean samples compared to noisy ones. Is this the case?

---

> > > ### Author Response · Authors · 2020-11-19
> > > **Thanks for your comments and recognition!**
> > >
> > > Thanks for your comments and recognition!
> > >
> > > **A1: More intuitions on $\alpha_{n,t}$ & $\ell_{CR}$**
> > >
> > > The high-level intuitions of specifying different $\alpha_{n,t}$s are primarily due to 1) the difficulty level of an example, and 2) the noise level in the example and the similar examples. Intuitively, when an instance is hard to be classified, and when the noise level is high and imbalanced (noise rates being substantially different for similar examples), we might want to customize the threshold for comparing to the prediction we are able to achieve on this instance. Similarly, for easy and clean examples, the range of feasible $\alpha$s might be much more available. Theoretically speaking,  setting $\alpha_{n,t}$ for each example provides a higher degree-of-freedom. But of course it becomes harder to identify them correctly. This can also be viewed as a technical contribution of the sample sieve.
> > >
> > >
> > > Indeed, the values of $\ell_{\text{CR}}(\bar{f}^{(t)}(x))$ and $\ell_{\text{CR}}(f^{(t)}(x))$ are the same. We use $\bar{f}^{(t)}(x)$ to distinguish from $f^{(t)}(x)$ that $\bar{f}^{(t)}(x)$ is the current/"old" model that will not contribute to back-propagation. We kept both $\bar{f}^{(t)}(x)$ and $f^{(t)}(x)$ here to be consistent with our sample sieve formulation in (2) (particularly, the objective in (2)). But we agree, when calculating $v_n$ at each iteration with a specific $f^{(t)}$, the two $\ell_{\text{CR}}$ terms can be canceled.  We clarified this technicality issue in bullet point \#3 below Figure 1 \& will further clarify after Eqn. (4).
> > >
> > > **A2: More intuitions on CR**
> > >
> > > We completely agree with the reviewer's intuition on why CR would be beneficial. Generally speaking (also what we observe in the experiment), if the rate of increase in confidence during training is faster for corrupted examples compared to clean ones, this means the correlation between features and corrupted labels is easier to be learned by NN.
> > >
> > > We'd like to clarify that we do have observations that NNs tend to learn the clean examples first before overfitting to noisy labels with instance-independent noise, but this observation becomes less clear when we have instance-dependent noise. For the instance-dependent label noise, as shown in Figure 2(e) and Figure 2(f), the NN with the CE loss is hard to provide a good division, indicating it learns clean examples almost along with corrupted examples. For CR, in Figure 2(g) and Figure 2(h), the division is better in the early stage (epoch-20), but not as good as the instance-independent setting. So our conjecture is that CR is regularizing NN to slow down learning/memorizing noisy examples (makes it hard for noisy labels to reduce the confidence in the prediction).
> > >
> > >
> > > We also want to acknowledge that the above property is also due to the assumption that the overall label information is dominantly informative that $T_{ii}(X) > T_{ij}(X)$. The assumption helps guarantee that with proper regulation, NN will receive more correct information statistically. Papers studying Coherent Gradients have similar intuitions and observations [R1, R2].
> > >
> > > We again want to thank the reviewer for pointing this out and we have added this at the bottom of page 3 in the revised version to better explain why CR works.
> > >
> > >
> > > [R1] Chatterjee S. Coherent Gradients: An Approach to Understanding Generalization in Gradient Descent-based Optimization. ICLR 2020.
> > >
> > > [R2] Zielinski P, Krishnan S, Chatterjee S. Explaining Memorization and Generalization: A Large-Scale Study with Coherent Gradients. arXiv preprint arXiv:2003.07422. 2020.

---

> ### Author Response · Authors · 2020-11-14
> **Technical contributions compared with peer loss (part2)**
>
> **A2: Technical contributions compared with peer loss**
> Our confidence regularization term is motivated by peer loss, but the rediscovered similar form offers us newly observed properties and theoretical guarantees. There are three main technical contributions compared with peer loss:
>
> i. *Novel confidence regularizer*
> Different from the perspective of peer loss that evaluates $ \ell (f (x,\tilde{y}) - \ell ( f(x\_{p\_1}), \tilde{y}\_{p\_2}) $ (where $p_1,p_2$ are two randomly sampled peer samples as named in the peer loss paper) jointly, our $\ell_{CR}$ term (analogous to $\ell(f(x_{p_1}),\tilde{y}_{p_2})$) does not require sampling peer samples, and we show that the form $\ell_{CR}$ alone is effectively serving as a regularizer that encourages confident predictions as proved in Theorem 1. This property has not been discovered or studied in peer loss and our reformulation helps partially explain the improved prediction performance in peer loss. We believe finding and proving this property is novel to the literature.
>
> ii. *Robust to instance-dependent label noise*
> Peer loss only guarantees the performance in binary classifications and particular multi-class extensions when $T_{ij} = T_{kj}, \forall j\ne k\ne i$. Technically, the proof of peer loss cannot handle the challenging instance-dependent label noise due to the important requirement on $T$ mentioned above. However, under mild assumptions, we prove the regularized loss (CE+CR, with properly tuned regularization parameter) is robust to instance-dependent label noise (Theorem 4, Contribution-1).
> Achieving the above result is primarily due to a novel analysis that decouples the effect of instance-dependent label noise with the regularized loss. We (theoretically) separate the regularized loss into three additive terms as in Theorem 3, which is novel and promising for understanding and controlling the effects of general instance-dependent label noise (Contribution-5). We believe this decoupled structure contributes to the literature and complements the peer loss work.
>
> iii. *Dynamic sieving solution*
> The confidence regularization property of CR enables us to design a theoretically guaranteed dynamic sieving solution, which is entirely new compared with peer loss. It is also not a trivial combination of peer loss and the "sample sieving" step as our dynamic sieving solution requires careful treatment and analysis of threshold $\alpha_n$ for each example. Both the regularized loss and the dynamic sieving solution are necessary components to guarantee the performance of the sample sieve. See comparison with GCE and other sample selection algorithms below for more details.

---

> ### Author Response · Authors · 2020-11-14
> **Technical contributions compared with GCE & intuitive explanations（part3）**
>
> **A3: Technical contributions compared with GCE**
>
> Comparing our sample sieve to the one used in GCE [2], we find two major differences: different loss functions and different thresholds.
>
> First of all, the robustness of GCE is not analyzed nor theoretically guaranteed when facing an instance-dependent noisy dataset. From empirical results, the bounded loss functions as GCE can improve the numerical stability of loss functions and thus are more robust to outliers than CE. Unlike the GCE loss, our regularized loss, when minimized over the instance-dependent noisy label, is guaranteed to find the classifier that best fits the underlying clean distribution in expectation with a properly balanced regularizer (Theorem 4). Therefore we think our loss works quite differently from GCE. It is critical to ensure the model is on the right direction of finding the clean distribution instead of fitting to a noisy one, otherwise we will not be granted the quality guarantee of our sample sieve procedure (Corollary 1). Recall the pure GCE loss (without sample selection) is not guaranteed to learn the clean distribution thus it is unclear of the quality of sieved examples if we substitute the GCE loss for our regularized loss.
>
> Additionally, the critical threshold for determining whether the example is clean or corrupted is set empirically and manually as a constant $k$ in GCE [2]. Theoretically, in the challenging instance-dependent label noise, the constant $k$ should be relevant to $T(X)$ as previously proved by Cheng et al. (2020). As shown in Section 2.2, the closed-form thresholds $\alpha_{n,t}$ in our sample sieve are specially designed for each example and calculated based on the model predictions with theoretical guarantees (Theorem 2). Getting $\alpha_{n,t}$ does not rely on $T(X)$. The difference between our solution and the previous ones (GCE, Co-teaching etc) can be viewed as being primarily due to we following different design philosophies toward sample selection. The sample selections in GCE, Co-teaching (Han et al., 2018), Co-teaching+ (Yu et al., 2019), JoCoR (Wei et al., 2020), and distillation  (Cheng et al. 2020) are designed to "pick up" clean examples while our approach is essentially "ruling out" corrupted examples (guaranteed in Theorem 2). Detailed comparisons of both philosophies are available in the feedback A2 to Reviewer-4.
>
>
> In the initial submission, we compared the performance of these works (except for Cheng et al. (2020) which focuses on binary classifications) and found them work well with instance-independent label noise but not well when facing instance-dependent noise. See Figure 3, Table 1, and Table 2 for results. Note the method $L_q$ (Zhang & Sabuncu, 2018) stands for GCE. We commented on this line of works including Co-teaching (Han et al., 2018), Co-teaching+ (Yu et al., 2019) and JoCoR (Wei et al., 2020) in Section 2.2 of the original submission. We have added comparisons with GCE and highlighted the difference in designing philosophies at the bottom of page 4 in the revised version. Thanks for your suggestion.
>
>
> **A4: Intuitive explanations**
>
> Intuitively, the sieving solution in GCE relies on a single threshold $k$ for the whole dataset while our sieving solution has closed-form thresholds $\alpha_{n}$ for each example. For different examples with different "pace" of being learned, it is hard to separate all the clean and corrupted examples perfectly with one threshold. With instance-dependent label noise, we empirically found that GCE might drop some critical information in those clean examples which are wrongly identified as being corrupted, while our sample sieve has guarantees to keep all the clean examples (Theorem 2) and the remaining examples after each iteration of sieve have non-increasing overall noise rates (more details can be found in responses A1 and A2 to Reviewer-4). Therefore, we shall believe our sample sieve approach is a more principled solution than GCE for instance-dependent label noise. We have shown intuitions at the bottom of page 4 in the revised version. Thanks for the suggestion to improve this paper.

---

> ### Author Response · Authors · 2020-11-14
> **DNNs with the CE loss overfit randomly assigned labels & fit clean distributions if CR is applied (part 4)**
>
> **A5: DNNs with the CE loss overfit randomly assigned labels & fit clean distributions if CR is applied**
>
> *Short response:*
> We totally agree that deep NNs can even perfectly overfit to datasets with randomly assigned labels, so as the noisy labels. Encouraging confident predictions (in the right direction) helps the model avoid overfitting to the noisy labels, which is exactly our motivation and inspiration of the confidence regularized approach.
>
> *Complete response:*
> Experimentally, it is true that deep networks can easily overfit to noisy labels [R5]. However, [R6] points out that even though the network fits random labels in the end, in early training stage, the network inclines to learning the right and clean distributions. This observation forms the basis of many noisy label learning methods, e.g., [R7], MentorNet (Jiang et al., 2017), Co-teaching (Han et al., 2018), M-correction (Arazo et al., 2019), Co-teaching+ (Yu et al., 2019). For example, Co-teaching (Han et al., 2018) selects clean examples at the beginning of training (since the network does not totally fit noisy labels at this time) and trains the network only on these examples. However, from Figure 3 in our initial submission, Co-teaching works well on symmetric and asymmetric noise settings and fails on instance-dependent settings whereas our methods still achieve higher performance. By comparing Figure 2(a) with Figure 2(e), we can infer this is because the instance-dependent label noise may be easier to be learned by DNNs than the instance-independent one. On the way from fitting clean labels to fitting noisy labels, the model predictions will be less confident (more stochastic). Our proposal of using the CR term to avoid fitting noisy labels by encouraging confident predictions at the early stage.
>
> Consider the case when the model perfectly learns (fits) the noisy distributions with the CE loss. In this case, with an infinite number of examples and enough model capacity, the model can perfectly learn the noisy distribution. For a particular feature $X$, if the ground-truth label is $1$ while the noisy dataset contains $(X,1)$ w.p. $0.6$ and $(X,0)$ w.p. $0.4$, the predicted probability would be $f_X[1] = 0.6$ and $f_X[0] = 0.4$. Compared to the training on clean datasets, label noise makes the model prediction less confident. That's why we mention "when model overfits to the noise, its predictions often become less confident" (wrt the true label distribution) in our initial submission.
>
>
> Learning the noisy distributions and processing separately using this distributional knowledge is another way to deal with the instance-dependent label noise as Cheng et al. (2020) while learning the unobserved clean distribution is also doable. Related works on "learning the clean distributions" include surrogate loss (Natarajan et al., 2013), DMI (Xu et al., 2019), and peer loss (Liu \& Guo, 2020). Note they all focused on the instance-independent label noise.
>
> We agree with the reviewer that "encouraging confidence makes the model harder to the noisy labels", which is exactly what we would like to achieve. Rather than overfitting to a noisy distribution confidently, we try to maintain a confident prediction towards the clean distribution given an instance-dependent noisy dataset. As shown in Theorem 4, with mild assumptions, minimizing the regularized loss is equivalent to minimizing the CE loss under the *clean distribution*. This is one major contribution of our method. We try to highlight this point in Contribution-1. In the revision, we revise our Contribution-1 to: "Rather than learn the noisy distribution, we propose to recover a classifier as if learning the clean distribution by a novel confidence regularization (CR) term and guarantee theoretically that ..."
>
> [R5] C. Zhang, S. Bengio, M. Hardt, B. Recht, and O. Vinyals. Understanding deep learning requires rethinking generalization. In ICLR, 2017.
>
> [R6] D. Arpit, S. Jastrzebski, N. Ballas, D. Krueger, E. Bengio, M. S. Kanwal, T. Maharaj, A. Fischer, A. Courville, Y. Bengio, et al. A closer look at memorization in deep networks. ICML, 2017.
>
> [R7] J. Huang, L. Qu, R. Jia, and B. Zhao. O2u-net: A simple noisy label detection approach for deep neural networks. ICCV 2019.

---

### Author Response · Authors · 2020-11-14
**Revised draft uploaded**

Dear reviewers and all,

We have revised our draft according to all the valuable comments. Major revisions are highlighted in blue. We sincerely thank all the reviewers. We would highly appreciate it if you could read our responses and revisions. Please feel free to let us know if further details/explanations would be helpful.

Best,

Authors

---

### Decision · Program_Chairs · 2021-01-07
**Final Decision**

**Decision:**

Accept (Poster)

**Comment:**

Dear Authors,

Thank you very much for your very detailed feedback to the reviewers. They have highly contributed to clarifying some of the concerns raised by the reviewers and improved their understanding of this paper.

Overall, all the reviewers acknowledge the merit of this paper and thus I suggest acceptance of this paper.
However, as Reviewer #4 pointed out, there are conceptual and theoretical issues that need to be more carefully addressed.
Please clarify these issues in the final version of the paper.